**Combining of Mie-Raman and fluorescence observations: a step forward in aerosol classification with lidar technology**

Igor Veselovskii[1], Qiaoyun Hu[2], Philippe Goloub[2], Thierry Podvin[2], Boris Barchunov[1], Mikhail Korenskii[1]

[1]*Prokhorov General Physics Institute of the Russian Academy of Sciences, Moscow, Russia.*

[2]*Univ. Lille, CNRS, UMR 8518 - LOA - Laboratoire d'Optique Atmosphérique, F-59650 Lille, France*

**Correspondence**: Qiaoyun Hu (qiaoyun.hu@univ-lille.fr)

**Abstract**

The paper presents an approach to reveal variability of aerosol type at high spatio-temporal resolution, by combining fluorescence and Mie-Raman lidar observations. The multi-wavelength Mie-Raman lidar system in operation at the ATOLL platform, Laboratoire d'Optique Atmosphérique, University of Lille, includes, since 2019, a wideband fluorescence channel allowing the derivation of the fluorescence backscattering coefficient $\beta_F$. The fluorescence capacity $G_F$, which is the ratio of $\beta_F$ to the aerosol backscattering coefficient, is an intensive particle property, strongly changing with aerosol type, thus providing a relevant basis for aerosol classification. In this first stage of research, only two intensive properties are used for classification: the particle depolarization ratio at 532 nm, $\delta_{532}$, and the fluorescence capacity, $G_F$. These properties are considered because they can be derived at high spatio-temporal resolution and are quite specific to each aerosol type. In particular, in this study, we use $\delta_{532}$ - $G_F$ diagram to identify smoke, dust, pollen and urban aerosol particles. We applied our new classification approach to lidar data obtained during 2020 – 2021 period, which includes strong smoke, dust and pollen episodes. The particle classification was performed with height resolution about 60 m and temporal resolution better than 8 minutes.

## 1. Introduction

Atmospheric aerosol is one of the key factors influencing the Earth's radiation budget through absorption and scattering of solar radiation and by affecting cloud formation. The processes of aerosol–radiation and aerosol-cloud interaction depend on aerosol size, shape, morphology,

absorption, solubility, etc., thus knowledge of the chemical composition and mixing state of the
aerosol particles is important for modeling of aerosol impact (Boucher et al., 2013). The aerosol
properties may vary in a wide range, so in practice usually several main types of aerosols are
separated on a base of their origin: e.g. urban, dust, marine, biomass burning (Dubovik et al., 2002).
Successful remote characterization of column integrated aerosol composition from the
observations of Sun – sky photometers and space-borne multiangle polarimeters was demonstrated
in numerous publications (Dubovik et al., 2002; Giles et al., 2012; Hamill et al., 2016; Schuster et
al., 2016; Li et al., 2019; Zhang et al., 2020). The aerosol impacts, however, depends also on
vertical variations/distributions of particle concentration and composition, which cannot be
derived from these instruments.

One of the recognized remote sensing techniques for vertical profiling of aerosol properties

is a lidar. Multiwavelength Mie-Raman and HSRL (High Spectral Resolution Lidar) lidar systems
provide unique opportunity to derive height-resolved particle intensive properties, such as lidar
ratios, Angstrom exponents and depolarization ratios at multiple wavelengths. Based on this
information, particle type can be determined (Burton et al., 2012, 2013; Groß et al., 2013; Mamouri
et al., 2017; Papagiannopoulos et al., 2018; Nicolae et al., 2018; Hara et al., 2018; Voudouri et al.,
2019; Wang et al., 2021; Mylonaki et al., 2021 and references therein). However, there is a
fundamental difference between particle classification based on the Sun – sky photometer and on
lidar observations. From both direct Sun and azimuth scanning measurements of the photometer
more than 100 observations are available. From this information the spectrally dependent
refractive index and absorption Angstrom exponent can be determined, which is important for
aerosol classification (Schuster et al., 2016; Li et al., 2019). The commonly used multiwavelength
lidars are based on a tripled Nd:YAG laser and are capable of providing three backscattering (355
nm, 532 nm, 1064 nm), two extinction (355 nm, 532 nm) coefficients and up to three particle
depolarization ratios (so called $3\beta+2\alpha+3\delta$ set). Thus the number of available lidar observations is
eight or less, which limits the performance of the aerosol typing algorithms. Nevertheless, the
results obtained by different research groups demonstrate that lidar-based particle identification is
possible. In publications of Burton et al. (2012, 2013) classification was performed from four
intensive parameters measured by the HSRL system: the lidar ratio at 532 nm ($S_{532}$), the
backscattering Angstrom exponent for 532/1064 nm wavelengths ($BAE_{532/1064}$), and particle
depolarization ratios at 532 nm and 1064 nm ($\delta_{532}$, and $\delta_{1064}$). With these input parameters eight
aerosol types: smoke, fresh smoke, urban, polluted maritime, maritime, dusty mix, pure dust and
ice were discriminated.
Important information on aerosol vertical distribution comes from the
EARLINET/ACTRIS lidar-network, aiming at unifying multiwavelength Mie-Raman lidar
systems over Europe (Pappalardo et al., 2014). For the automation of aerosol classification, several
approaches were developed in the frame of EARLINET. These approaches include the
Mahalanobis distance-based typing algorithm (Papagiannopoulos et al., 2018), a neural network
aerosol classification algorithm (NATALI) (Nicolae et al., 2018), and algorithm based on source
classification analysis (SCAN) (Mylonaki et al., 2021). All these algorithms have demonstrated
their ability for aerosol classification. In particular, the NATALI is able to identify up to 14 aerosol
mixtures from $3\beta+2\alpha+1\delta$ observations.
Nevertheless, the above-mentioned algorithms have to deal with a fundamental limitation:
the particle intensive properties, even for pure aerosols (generated by a single source) exhibit
strong variations. For example, the lidar ratio $S_{355}$ of smoke in publication of Nicolae et al. (2018)
varies in 38 sr – 70 sr range, and in our own measurements we observed for aged smoke $S_{355}$ as
low as 25 sr (Hu et al., 2021). Strong variation of smoke lidar ratios in EARLINET/ACTRIS
observations is discussed also in the recent publication of Adam et al. (2021). Such uncertainty in
parameters of the aerosol model complicates the aerosol classification. Thus, it is desirable to
combine the Mie-Raman observations with another range resolved technique, providing additional
independent information about aerosol composition. Such information can be obtained from laser
induced fluorescence emission.
Application of fluorescence lidar technique was intensively considered during the last
decade to study aerosol particles. Lidar measurements of the full fluorescence spectrum with
multianode photomultipliers (Sugimoto et al., 2012; Reichardt et al., 2014, 2017; Saito et al., 2022)
provides an obvious advantage in particle identification. However, even a more simple
fluorescence lidar with a single wideband fluorescence channel, opens new opportunities for
aerosol characterization (Veselovskii et al., 2021; 2022; Zhang et al., 2021). Such fluorescence
configuration could be implemented in existing Mie-Raman lidars, and the fluorescence
backscattering coefficient $\beta_F$ is calculated from the ratio of fluorescence and nitrogen Raman
signals. To characterize the aerosol fluorescence properties, the fluorescence capacity $G_F$ is
introduced as the ratio of $\beta_F$ to aerosol backscattering coefficient at one of laser wavelengths
(Veselovskii et al., 2020b). In this study, the backscattering at 532 nm was used. The fluorescence
capacity is an intensive particle parameter, which changes strongly with aerosol type, being the
highest for smoke and the lowest for dust. Thus, the combination of Mie – Raman and fluorescence
backscatter provides a basis to improve particle classification. A Mie – Raman lidar provides
several particle intensive parameters, however, the profiles of particle parameters associated with
the extinction coefficient, such as lidar ratio or extinction Angstrom exponent, may contain strong
noises, because the extinction coefficients are derived from the slope of Raman lidar signals, thus
averaging over significant spatio-temporal intervals is demanded. Meanwhile, the particle
depolarization and the fluorescence *capacity* can be calculated with high spatio-temporal
resolution.
Recently, we have demonstrated that the $\delta - G_F$ diagram allows to separate several aerosol
types, such as dust, pollen, urban (continental) and smoke (Veselovskii et al., 2021a). In the present
study, we use this technique to classify aerosol particle types in the troposphere at high spatio-
temporal resolution. We present results of aerosol classification on the basis of fluorescence and
Mie-Raman lidar measurements performed at the ATOLL (ATmospheric Observation at liLLe) at
Laboratoire d'Optique Atmosphérique, University of Lille, during 2020 – 2021 period, which
includes strong smoke, dust and pollen episodes. Paper starts with a description of the experimental
setup and data processing scheme in Sect.2. In Sect.3 we present the algorithm for aerosol
classification on a base of depolarization and fluorescence measurements. Results of the
application of the developed approach to different atmospheric situations, including smoke, dust
and pollen episodes are given in Sect.4.

**2.   Experimental setup and data analysis**
*2.1. Lidar system*
The multiwavelength Mie-Raman lidar LILAS (LIlle Lidar AtmosphereS) is based on a
tripled Nd:YAG laser with a 20 Hz repetition rate and pulse energy of 70 mJ at 355 nm.
Backscattered light is collected by a 40 cm aperture Newtonian telescope and the lidar signals are
digitized with Licel transient recorders with 7.5 m range resolution, allowing simultaneous
detection in the analog and photon counting mode. The system is designed for the detection of
elastic and Raman backscattering, allowing the so called $3\beta+2\alpha+3\delta$ data configuration, including
three particle backscattering ($\beta_{355}$, $\beta_{532}$, $\beta_{1064}$), two extinction ($\alpha_{355}$, $\alpha_{532}$) coefficients along with
three particle depolarization ratios ($\delta_{355}$, $\delta_{532}$, $\delta_{1064}$). The particle depolarization ratio, determined
as a ratio of cross- and co-polarized components of the particle backscattering coefficient, was
calculated and calibrated in the same way as described in Freudenthaler et al. (2009). Many
calibration and operation procedures have been automated for the LILAS system to improve the
overall performance of the lidar in terms of observation frequency and data quality.  The aerosol
extinction and backscattering coefficients at 355 and 532 nm were calculated from Mie-Raman
observations (Ansmann et al., 1992), while $\beta_{1064}$ was derived by the Klett method (Klett, 1985).
The full geometrical overlap was achieved at approximately 750 m range. For calculation of $\alpha$ and
$\beta$ at 532 nm we use the rotational Raman scattering instead of the vibrational one (Veselovskii et
al., 2015), which allows to increase the power of Raman backscatter and to decrease separation
between the wavelengths of elastic and Raman components. Additional information about
atmospheric parameters was available from radiosonde measurements performed at Herstmonceux
(UK) and Beauvechain (Belgium) stations, located 160 km and 80 km away from the observation
site respectively.

The LILAS system can also profile the laser induced fluorescence of aerosol particles. A

part of the fluorescence spectrum is selected by a wideband interference filter of 44 nm width
centered at 466 nm. The strong sunlight background at daytime restricts the fluorescence
observations to nighttime hours. The fluorescence backscattering coefficient $\beta_F$, is calculated from
the ratio of fluorescence and nitrogen Raman backscattering signal, as described in Veselovskii et
al. (2020b). This approach allows us to evaluate the absolute values of $\beta_F$, if the relative sensitivity
of the channels is calibrated and the nitrogen Raman scattering differential cross section is known.
All $\beta_F$ profiles presented in this work were smoothed with the Savitzky – Golay method, using
second order polynomials with 21 points in the window. For the calculation of the fluorescence
capacity $G_F$, in principle, backscattering coefficients at any laser wavelength can be used. In our
study we always used $\beta_{532}$, because it is calculated with the use of rotational Raman component
and is considered to be the most reliable, thus the fluorescence capacity is calculated as $G_F = \dfrac{\beta_F}{\beta_{532}}$.
*2.2.Calculation of the particle backscattering coefficient from Mie-Raman measurements*

Mie – Raman lidar measurements allow independent evaluation of aerosol extinction and

backscattering coefficients. Commonly used approach for $\beta$ calculation was formulated in the
paper of Ansmann et al. (1992). This approach includes the choice of a reference height, where the
scattering is purely molecular. However, such height range is not always available, for example,
in the presence of the low level clouds. Moreover, when long-term spatio-temporal variations of
backscattering coefficients are analyzed, the uncertainty in the choice of the reference height leads
to oscillations in $\beta$ profiles. To resolve this issue, we modified the Raman method as described
below.
In an elastic channel, the backscattered radiative power $P_L$, at wavelength $\lambda_0$ and distance
$z$ is described by the lidar equation:
$$P_L = O(z)\frac{1}{z^2}C_L(\beta_L^a + \beta_L^m)\exp\left\{-2\int_0^z(\alpha_L^a + \alpha_L^m)dz'\right\} = O(z)\frac{1}{z^2}C_L(\beta_L^a + \beta_L^m)T_L^2, \tag{1}$$
while in a Raman channel, it can be written as:
$$P_R = O(z)\frac{1}{z^2}C_R\beta_R\exp\left\{-\int_0^z(\alpha_L^a + \alpha_R^a + \alpha_L^m + \alpha_R^m)dz'\right\} = O(z)\frac{1}{z^2}C_R\beta_R T_L T_R. \tag{2}$$
Here $O(z)$ is the geometrical overlap factor, which is assumed to be the same for elastic and Raman
channels. $C_L$ and $C_R$ are the range independent constants, including efficiency of the detection
channel. $T_L$ and $T_R$ are one-way transmissions, describing light losses on the way from the lidar to
distance $z$ at laser $\lambda_L$ and Raman $\lambda_R$ wavelengths. Backscattering and extinction coefficients contain
aerosol and molecular contributions: $\beta_L^a + \beta_L^m$ and $\alpha_L^a + \alpha_L^m$, where the superscripts "$a$" and "$m$"
indicate aerosol and molecular scattering, respectively. Raman backscattering coefficient is:
$$\beta_R = N\sigma_R, \tag{3}$$
where $N$ is the number of Raman scatters (per unit of volume) and $\sigma_R$ is the Raman differential
scattering cross section in the backward direction.
Dividing equation (1) on (2) we get:
$$\frac{P_L}{P_R} = \frac{C_L}{C_R}\frac{(\beta_L^a + \beta_L^m)}{\beta_R}\frac{T_L}{T_R} \tag{4}$$
Backscattering coefficient is calculated from (3) and (4) as:
$$\beta_L^a = \frac{P_L}{P_R}\frac{C_R}{C_L}\sigma_R N\frac{T_R}{T_L} - \beta_L^m = \frac{P_L}{P_R}KN\frac{T_R}{T_L} - \beta_L^m \tag{5}$$
The differential transmission $\dfrac{T_L}{T_R}$ can be calculated the same way, as it is done for the water vapor
measurements (Whiteman, 2003). For rotational Raman signal, which we use in our 532 nm
channel (Veselovskii et al., 2015), $\lambda_L \approx \lambda_R$, so $\dfrac{T_L}{T_R} = 1$.
The calibration constant $K = \dfrac{C_R}{C_L}\sigma_R$ can be found by comparing $\beta_L^a$ in Eq.5 with the
backscattering coefficient $\tilde{\beta}_L^a$ computed with the traditional Raman method, using the reference
height (Ansmann et al., 1992).
$\quad K = (\tilde{\beta}_L^a + \beta_L^m)\dfrac{P_R}{P_L}\dfrac{1}{N}\dfrac{T_L}{T_R}$ (6)
For simplicity, hereinafter we will use notation $\beta_L$ instead $\beta_L^a$. Thus, if during the measurement
session we have a temporal interval, where the reference height is available, we can determine the
calibration constant $K$ and use it for $\beta_L$ calculations from eq.5, assuming that relative sensitivity of
channels during the session is not changed. Even if cloud layers occur during the whole session,
we can use $K$ from the previous cloud-free profiles (assuming, again, that the relative sensitivity
of channels is the same). We will call this approach for $\beta$ calculation as "modified Raman method",
to distinguish it from traditional one (Ansmann et al., 1992).
To estimate variations of the relative sensitivity of the channels, we analyzed long-term
cloudless measurements when the reference height was available for every individual profile. The
results demonstrate that variations of calibration constant during the session (about 8 hours) were
below 3%. Fig.1 and 2 present the application of this modified Raman method to the measurements
on 2 March 2021. The dust layer extended from 2 km to 8 km height and inside this layer the ice
and liquid clouds were formed during the 00:00 – 05:00 UTC interval, thus $\beta_{532}$ could not be
calculated with traditional Raman technique. The temporal interval 19:00 – 20:00 was used to find
calibration constant $K$. Fig.1 shows vertical profiles of backscattering coefficient $\tilde{\beta}_{532}$ calculated
with traditional Raman method (with reference height), and $\beta_{532}$ calculated with modified method
(with the calibration constant). Profiles of $\tilde{\beta}_{532}$ and $\beta_{532}$ coincide for the whole height range. The
calibration constant $K$, shown on the same plot, does not demonstrate height dependence, though
oscillations around the mean value increase with height. For computations, we choose the value of
$K$ at low altitudes averaged inside some height interval.
Fig.2 provides spatio-temporal variations of $\beta_{532}$, particle depolarization $\delta_{532}$ and the
fluorescence capacity $G_F$. Depolarization measurements reveal the presence of dust ($\delta_{532} \approx 30\%$)
and the ice cloud above 4 km ($\delta_{532} > 40\%$). The liquid cloud below 4 km after midnight can be
identified by a low depolarization ratio $\delta_{532} < 3\%$. The fluorescence capacity of dust is low, about
$0.2 \times 10^{-4}$. However, below 2 km, $G_F$ is significantly higher, up to $1.2 \times 10^{-4}$. In combination with a
high depolarization ratio (up to 20%), it can indicate the presence of pollen at low altitudes. On
the fluorescence capacity panel, we can see that after 01:00 UTC the dust and pollen layers are
mixed below 2 km, resulting in a value of $G_F$ about $0.5 \times 10^{-4}$. The fluorescence capacity inside ice
and liquid clouds is below $0.01 \times 10^{-4}$. Fig.2 clearly demonstrates the advantage of simultaneous
depolarization and fluorescence measurements for the study of cloud formation in the presence of
aerosol. All spatio-temporal distributions of $\beta_{532}$ presented in this paper were calculated from Eq.5
with a modified Raman method.

**3. Aerosol classification based on fluorescence measurements**
**3.1.** *Approach for aerosol classification*.
As was discussed in our recent publication (Veselovskii et al., 2021), the $\delta$-$G_F$ diagram
allows to separate several aerosol types, including smoke, dust, pollen, urban, ice and liquid water
particles. Smoke and urban aerosols both have a small depolarization ratio, but the fluorescence
capacity of smoke is almost one order higher, so these particles can be separated. Dust and pollen
both have high depolarization ratio (up to 30%), but $G_F$ of dust is significantly lower, which again
provides basis for discrimination. The depolarization ratio of some aerosol types is characterized
by strong spectral dependence. For example, the depolarization ratio of aged smoke decreases with
wavelength. It is below 5% at 1064 nm but at 355 nm in upper troposphere it may exceed 20%
(Burton et al., 2015; Haarig et al., 2018; Hu et al., 2019; Veselovskii et al., 2022), which
complicates smoke and dust separation. For pollen, on the contrary, the depolarization ratio at
1064 nm can be the highest (Veselovskii et al., 2021). Thus, choice of $\delta_{1064}$ for $\delta$-$G_F$ diagram could
be advantageous. However, as mentioned, the backscattering coefficient at 1064 nm is calculated
with Klett method (Klett, 1985), which, besides assumption about lidar ratio, needs reference
height and cannot be used in cloudy situations. This is why in our study we used the $\delta_{532}$-$G_F$
diagram.

In our present work, we consider a simple classification scheme since we use only two
intensive parameters $G_F$ and $\delta_{532}$. Our goal is to demonstrate that in the $\delta_{532}$-$G_F$ diagram, our lidar
observations form clusters and characteristic patterns which can be attributed to different aerosol
types or their mixtures. We consider four aerosol types: dust, smoke, pollen and urban, and two
cloud types: liquid and ice clouds. Dust and pollen are large particles of complicated shape,
characterized by high depolarization ratio, while smoke and urban pollution are small particles
with low depolarization. In our classification "urban aerosol" includes continental aerosol, sulfates
and soot. At this stage, we do not yet consider absorption to discriminate particles.

The choice of the range of particle properties variation for each aerosol type is an important
aspect of the approach. Typical ranges of $G_F$ and $\delta_{532}$ variations used in our classification scheme
are given in Table 1 and are shown in Fig.3. These ranges are based on results obtained in LOA
(Laboratoire d'Optique Atmosphérique) and on results presented in aerosol classification studies
(Burton et al., 2012, 2013; Nicolae et al., 2018; Papagiannopoulos et al., 2018, Mylonaki et al.,
2021).

**Dust**. The depolarization ratio, $\delta_{532}$, of Saharan dust near the source regions is up to 35%
(Veselovskii et al., 2020a). However, after transportation and mixing with local aerosol $\delta_{532}$ can
be as low as 20% (Rittmeister et al., 2017). In many studies, the dust events having with smaller
depolarization ratio are classified as "polluted dust" (e.g. Burton et al., 2012, 2013). At the moment,
we do not introduce the discrimination between the two subtypes and mark as "dust" the particles
with $20\% < \delta_{532} < 35\%$, and $0.1 \times 10^{-4} < G_F < 0.5 \times 10^{-4}$.

**Smoke**. In 2021-2022, we regularly observed, over ATOLL platform, smoke layers
originated from Californian and Canadian forest fires (Hu et al., 2022). The particle depolarization
and fluorescence capacity of this transported smoke varied from episode to episode and, for
classification, we selected the ranges $2\% < \delta_{532} < 10\%$, $2 \times 10^{-4} < G_F < 6 \times 10^{-4}$. At this stage, we do not
discriminate "fresh" and "aged" smoke, and the range of $\delta_{532}$ variation is similar to the one used
in classification of Burton et al. (2012).

**Pollen**. The pollen over north of France is usually mixed with other aerosol and the
particles which we mark as "pollen" are actually the mixtures. Depolarization ratio of clean pollen
varies strongly for different taxa. For birch pollen, Cao et al. (2010) reported $\delta_{532}=33\%$, and in the
measurements over Finland during birch pollination (Bohlmann et al., 2019), observed values of
$\delta_{532}$ up to 26%. The observations over Lille during pollen season (Veselovskii et al., 2021a) rarely
revealed values $\delta_{532}$ exceeding 20%. Based on that observations, we type as "pollen" the particles
mixtures with $15\%<\delta_{532}<30\%$, and $0.8\times10^{-4}<G_F<3.0\times10^{-4}$.

**Urban**. This type of aerosol includes a variety of particle types (e.g. sulfates, soot) and its

properties may depend on the relative humidity. Based on our measurements inside the boundary
layer, for classification we choose the ranges $1\%<\delta_{532}<10\%$, and $0.1\times10^{-4}<G_F<1.0\times10^{-4}$. Similar
range for $\delta_{532}$ is used in classification of Burton et al. (2013). Urban and smoke particles both have
a low depolarization, but the smoke fluorescence capacity can be up to one order higher, so these
particles can be reliably discriminated.

**Ice and water clouds**. Both cloud types have low fluorescence capacity $G_F<0.01\times10^{-4}$.

However, the ice clouds are usually observed at the heights, where fluorescence signal is low and
can not be used for classification. Thus above ~8 km, the ice cloud are identified by high
depolarization ratio $\delta_{532}>40\%$. Depolarization ratio of the liquid water clouds is usually affected
by the effects of the multiple scattering, so for their identification we use $\delta_{532}<5\%$.

The analysis of aerosol mixtures is an important subject and, the possibility to separate the

mixture components based on lidar measurements was discussed in publications of Sugimoto and
Lee (2006), Gross et al. (2011), Gasteiger et al. (2011), Tesche et al. (2009), Burton et al. (2014).
The information about mixture composition can be also revealed in $\delta_{532}$-$G_F$ diagram. For example,
pollen can be mixed with urban particles. At different heights the pollen contributes differently to
$\beta_{532}$, so at $\delta_{532}$-$G_F$ diagram, the data points will form the pattern, which extends from location,
attributed to "pure" urban aerosol to location, attributed to "pure" pollen. To estimate, how such
pattern looks like, a simplified modeling for fixed particle parameters was performed.
Corresponding results are shown in Fig.3 by symbols (circles). The particle depolarization ratio $\delta$
of the mixture, containing urban aerosol (*u*) and pollen (*p*), with depolarization ratios $\delta^u$ and $\delta^p$,
can be calculated as:
$$\delta = \frac{\left(\dfrac{\delta^p}{1+\delta^p}\right)\beta^p + \left(\dfrac{\delta^u}{1+\delta^u}\right)\beta^u}{\dfrac{\beta^p}{1+\delta^p} + \dfrac{\beta^u}{1+\delta^u}} \tag{7}$$

The fluorescence capacity of the mixture is given by:
$$G_F = \frac{\beta^u G_F^u + \beta^p G_F^p}{\beta}$$  (8)
Here total backscattering $\beta = \beta^u + \beta^p$.
The computations in Fig.3 were performed for values of pollen contribution $\dfrac{\beta_{532}^p}{\beta_{532}}$ in 0 - 1.0
range with step 0.1. We assume that the depolarization ratios of pollen and urban aerosol are $\delta_{532}^p$
=30% and $\delta_{532}^u$=3%, while the fluorescence capacities are $G_F^u$=0.2×10$^{-4}$ and $G_F^p$=2.5×10$^{-4}$. We
remind that the fluorescence capacities are calculated at 532 nm wavelength. In the $\delta_{532}$-$G_F$
diagram the computed points provide a characteristic curve, which in the next section will be
compared with experimental results. The same computations were performed for a smoke (s) and
dust (d) mixture, assuming $\delta_{532}^d$=30%, $\delta_{532}^s$=3%, $G_F^d$=0.2×10$^{-4}$ and $G_F^s$=4.0×10$^{-4.}$ Corresponding
results are shown in Fig.3 with stars. In a similar way, the characteristic curves for other mixtures
can be also represented.
We are also able to identify liquid water and ice layers. Liquid water cloud layers have low
fluorescence capacity ($G_F$<0.01×10$^{-4}$) and $\delta_{532}$<3%. Ice particles also have low $G_F$, but at heights
where ice clouds are usually observed, the signal of fluorescence backscattering is noisy. Thus at
high altitudes ice particles are discriminated by a high depolarization ratio $\delta_{532}$>40 %.

**3.2. *Classification of spatio-temporal observations***
The input parameters in our classification scheme are the spatio-temporal distributions of $\beta_{532}$,
$\delta_{532}$ and $G_F$, which are presented as matrices $\beta_{532}^{i,j}$, $\delta_{532}^{i,j}$, $G_F^{i,j}$, where $i$=1… $N_T$; $j$=1… $N_H$. Values
$N_T$ and $N_H$ are the numbers of temporal and height intervals in the analyzed dataset. In a single
measurement we accumulate 2×10$^3$ laser pulses, so temporal resolution of the measurements is
about 100 s, while the height resolution is 7.5 m.
The particle intensive properties cannot be evaluated reliably when the backscattering
coefficient is low. Thus, we set a threshold value for $\beta_{532}$ (normally 0.2 Mm$^{-1}$sr$^{-1}$); namely, when
$\beta_{532}^{i,j}$< 0.2 Mm$^{-1}$sr$^{-1}$ the elements of the matrices $\delta_{532}^{i,j}$ and $G_F^{i,j}$, are classified as "low signal" and
ignored. For the remaining elements, we determine the aerosol type, using our approach. A primary
typing is being made for each point *(i,j)* separately, in accordance with parameter ranges given in
the Table 1. The elements, which are out of all these ranges, are marked as "undefined". We
consider 6 types of the particles, respectively dust, smoke, pollen, urban, ice crystals and water
droplets. Moreover, there can be two additional results of primary typing: "undefined" and "low
signal". Thus, there are altogether 8 possible results of primary typing. For every aerosol type, a
$N_T \times N_H$ dimension matrix is constructed. If at this first stage of classification some single pixel
point (i, j) is classified as, e.g., dust, the corresponding value in the 'dust' matrix is set to 1,
otherwise it is set to 0.

The single pixel particle parameters contain statistical noise, which influences the results of

the primary typing, thus producing high frequency oscillations of non-physical character. From a
physical point of view, the aerosol single-type areas should form smooth regions, so a special
smoothing procedure (stage 2 of our algorithm) was developed to remove the oscillations. The
smoothing procedure is based on a convolution with Gaussian kernel

$$Z = \exp\left( -\left( \frac{t^2}{s_T^2} + \frac{h^2}{s_H^2} \right) \right)$$

(9)

where $t$ and $h$ are temporal and height coordinates. The resolution of typing is being controlled by
the parameters $s_T$ and $s_H$, which are set as the number of temporal and height bins.

On the second stage of classification each of these matrices is separately convoluted with

the Gauss kernel Z. After the convolution, the values for each pixel (i,j) are being compared. If,
e.g., the 'dust' matrix contains maximal value at the pixel (i,j), in respect to all other matrices, then
the pixel (i,j) is finally classified as dust. The choice of smoothing parameters depends on aerosol
loading and aerosol type. For the measurements inside the boundary layer in many cases the single
pixel typing ($s_T$=1, $s_H$=1) is possible, while for analysis of the weak elevated layers the smoothing
should be applied. All results presented in this study were obtained for $s_T$=3 and $s_H$=5, thus the
temporal and range resolutions of our typing procedure are estimated to be about 8 minutes and
60 m respectively.

**4.  *Application of classification approach to LILAS data***

The classification approach, described in the previous section, was applied to the data of

the Mie-Raman- Fluorescence lidar at the ATOLL platform, located on the campus of Lille
University, during 2020 – 2021 period. Here we present results of aerosol classification for several
relevant atmospheric situations, to demonstrate that different aerosol types are well separated
based on $\delta_{532}$-$G_F$ diagram.

### *12 September 2020: Wildfire smoke*

Fig.4 presents the spatio-temporal variations of aerosol and fluorescence backscattering
coefficients ($\beta_{532}$ and $\beta_F$) together with the particle depolarization ratio $\delta_{532}$ and the fluorescence
capacity $G_F$ during smoke episode on the night 12-13 September 2020. The smoke layer extends
from approximately 2 km to 5 km height, and it is characterized by high fluorescence capacity
$G_F>3.0\times10^{-4}$ and low depolarization ratio $\delta_{532}<7\%$. The cirrus clouds occurred above 11 km height
during the whole night. The smoke layer was transported from North America; detailed analysis
of the layer origin and transportation is given in the recent publication of Hu et al. (2022). The
results of aerosol typing for this episode are shown in Fig.5. On the $\delta_{532}$-$G_F$ diagram these data
form two clusters. First cluster includes points in the range $2.0\times10^{-4}<G_F<6.0\times10^{-4}$ and
$2\%<\delta_{532}<7\%$, such high fluorescence and low depolarization should be attributed to smoke
particles. The second cluster consists of points localized inside $0.1\times10^{-4}<G_F<0.8\times10^{-4}$ and
$1\%<\delta_{532}<3\%$ intervals which corresponds to urban particles in Table 1. After cluster localization,
the observations can be plotted as aerosol types, using the parameters in Table 1 and the approach,
described in section 3.2. The aerosol types in Fig.5b are spatially separated and contain no high
frequency oscillations. Urban particles are localized at low heights, below 1 km. We would like to
remind that, at the condition of high relative humidity (RH), the fluorescence capacity can decrease
due to the particle's hygroscopic growth. The water uptake increases the particle backscattering,
but does not change the fluorescence. As a result, the fluorescence capacity decreases. (Veselovskii
et al., 2020). In accordance with radiosonde data the relative humidity below 1 km was quite high
(about 70% at 500 m) and decreased with height, which can explain the wide range of $G_F$ variation
observed for urban particles in Fig.5a.
The particle intensive properties, such as the lidar ratios at 355 nm and 532 nm wavelengths
($S_{355}$, $S_{532}$), the particle depolarization ratios ($\delta_{355}$, $\delta_{532}$, $\delta_{1064}$), the extinction ($A^{\alpha}_{355/532}$) and the
backscattering ($A^{\beta}_{355/532}$, $A^{\beta}_{532/1064}$) Angstrom exponents for the episodes analyzed in this study, are
summarized in Table 2. For this measurement session, in the smoke layer the lidar ratio at 532 nm
significantly exceeds corresponding value at 355 nm ($S_{532}=80\pm12$ sr and $S_{355}=50\pm7$  sr). The
particle depolarization ratio decreases with wavelength from 4.5% at 355 nm to 2% at 1064 nm.
Such spectral dependence of the lidar ratio and depolarization ratio for the aged smoke is in
agreement with previous studies (e.g. Haarig et al., 2018; Hu et al., 2022 and references therein).

*30 May 2020: Urban vs Pollen*

Pollen grains represent a significant fraction of primary biological materials in the
troposphere and fluorescence induced emission provides an opportunity for their identification.
Fig.6 presents spatio-temporal variations of $\beta_{532}$, $\beta_F$, $\delta_{532}$, $G_F$ during pollen season on the night 30-
31 May 2020. Presence of different types of pollen over Lille in Spring – Summer 2020 was
discussed in our recent publication (Veselovskii et al., 2021). In particular, on 30 May 2020 the in
situ measurements at the roof of the building demonstrate the presence of significant amount of
grass pollen. The transport of pollen can be analyzed with a global-to-meso-scale dispersion model
SILAM (Sofiev et al., 2015). In Appendix we show the maps of the pollen index, for four sessions
from this study at 22 UTC. On 30 May the pollen index in Lille region is about 5.0, indicating high
content of pollen.
The aerosol is located inside the planetary boundary layer (PBL) below 2.5 km. At altitudes
below 1 km, the depolarization ratio $\delta_{532}$ after 23:00 increases up to ~15% simultaneously with an
increase of the fluorescence capacity up to $2.0\times10^{-4}$, which can be an indication of pollen presence.
On the $\delta_{532}$-$G_F$ diagram in Fig.7a, the single pixel data points spread from the values typical for
the urban particles to the values typical for the pollen. Contribution of pollen to the total
backscattering changes with height and the points form the pattern, similar to characteristic curve,
calculated for urban – pollen mixture in Fig.3. In accordance with radiosonde data from
Herstmonceux station, the RH at midnight was about 40% at 500 m and it increased up to 70% at
2000 m, thus the spatio – temporal variations of RH could influence the observed values of the
backscattering coefficient and depolarization ratio. In particular, the hygroscopic growth can
decrease the values of both $\delta_{532}$ and $G_F$. However, the value of the fluorescence capacity in Fig.7a
changes for almost one order of magnitude, and such strong change in $G_F$ can not be explained by
the particle hygroscopic growth only. For example, from the recent publication of Sicard et al.
(2022), increase of $\beta_{532}$ of urban aerosol for this range of RH, is below the factor 1.5. Thus, we
suppose that the pattern in Fig.7a is due to the mixing urban and pollen particles The spatio–
temporal distribution of aerosol types is shown in Fig.7b. The urban particles (brown) are
predominant, while pollen (yellow) is localized below 1 km height. The grey color corresponds to
unidentified aerosol type which, in our case, is the mixture of urban particles and pollen.
An indicator of pollen presence in an aerosol mixture, along with high depolarization ratio,
can be a higher value of $\delta_{1064}$ in respect to $\delta_{532}$ or $\delta_{355}$ (Cao et al., 2010; Veselovskii et al., 2021).
Vertical profiles of the particle depolarization ratio at all three wavelengths for this episode are
given in Fig.8c of Veselovskii et al. (2021). At 0.75 km height, where $\delta_{1064}$ is about 15%, the ratio
$\dfrac{\delta_{1064}}{\delta_{532}}$ is 1.5, which corroborates suggestions about pollen presence. For urban aerosol the
depolarization spectral ratio $\dfrac{\delta_{1064}}{\delta_{532}}$ can be also above 1.0 (Burton et al., 2013), but absolute values
of depolarization are significantly lower than for pollen particles (below 10%).

*14 September 2020: wildfire smoke vs pollen mixture*
Another strong smoke episode occurred in the night 14-15 September 2020, and
corresponding distributions of $\beta_{532}$, $\beta_F$, $\delta_{532}$, and $G_F$ are shown Fig.8. The elevated smoke layer
with low depolarization ratio ($\delta_{532}<5\%$) and high fluorescence capacity (up to $4.0\times10^{-4}$) was
observed at approximately 6 km height during the whole night. Inside the boundary layer the
depolarization ratio is higher, up to 15%, while fluorescence capacity is lower (about $1.0\times10^{-4}$),
compared to the elevated layer. On the $\delta_{532}$-$G_F$ diagram in Fig.9a we can see the cluster of data
points, corresponding to the smoke. The same time, a part of the points are inside the range of
parameters attributed to the pollen (Table 1). The remaining points should be attributed to the
mixture of pollen, smoke and urban aerosol. On the distribution of the particle types (Fig.9b) this
mixture is marked with gray color. The pollen particles are localized below 1 km. Presence of
pollen over Lille in September is not common, but it can be transported from other regions. The
SILAM pollen index in Fig.A1 for this date demonstrates the transport of pollen to northern France
from the southeast of France and the east Mediterranean.
Fig.10a presents profiles of $\delta_{532}$ and $\delta_{1064}$ together with $\beta_{532}$ for the temporal interval 00:00
– 04:00 UTC. The relative humidity, in accordance with radiosonde data from Herstmonceux
station, did not exceed 50% below 1.7 km. Above that height RH increased up to 75% at 2.5 km,
thus the observed increase of $\beta_{532}$ above 1.5 km can be partly related to RH growth. The relative
humidity inside the smoke layer did not exceed 10%. Similarly to Fig.8, $\delta_{1064}$ exceeds $\delta_{532}$ at low
heights. The ratio $\frac{\delta_{1064}}{\delta_{532}}$ is about 1.5 at 1 km and inside the smoke layer $\frac{\delta_{1064}}{\delta_{532}} \approx 0.4$. Higher values
of depolarization ratio at 532 nm compared to 1064 nm are reported for aged smoke by Haarig at
al. (2018), Hu et al. (2019, 2022). The BAE does not present significant height variations: $A^{\beta}_{532/1064}$
is about 1.0 inside the PBL and it increases to 1.25 inside the smoke layer (Fig.10b).
Simultaneously, the fluorescence capacity in the smoke layer increases about a factor 4, comparing
to the PBL, which demonstrates efficiency of the fluorescence technique for discriminating smoke
from other aerosol types.

*10 April 2020: Urban vs Pollen*

In the beginning of April, we experienced several atmospheric situations, for which

elevated layers were classified as urban aerosols. One of such cases, on the night 10 -11 April 2020,
is shown in Fig.11. Lidar observations were performed at an angle of 45 degrees to the horizontal,
so the minimum height reachable in the analysis is 350 m. The relative humidity, in accordance
with radiosonde data from Herstmonceux station, increased with height from 54% at 1.0 km to 65%
at 2.2 km. The layer with depolarization ratio $\delta_{532}$ below 5% was observed at about 2 km height
during the night. The fluorescence capacity in the layer is low (below $0.5\times10^{-4}$), so it is identified
as urban aerosol. HYSPLIT backward trajectories (not shown) indicate that the air masses at 750
m and 2000 m heights were transported from England (HYSPLIT, 2022). For the period 21:00 –
23:00 UTC the depolarization ratio below 500 m has increased simultaneously with the
fluorescence capacity, which can be an indication of pollen presence.

On the $\delta_{532}$-$G_F$ diagram (Fig.12a) the single pixel measurements in 350 m – 1500 m and

1500 m – 2500 m height ranges are shown by different colors. The data points related to the upper
layer are within the range of parameters expected for urban aerosol. The points in the lower layer
(below 1500 m), are partly out of this range, so the aerosol type for these points is undefined. We
assume that this is the mixture of urban and pollen particles, because we observe particles with
high depolarization ($\delta_{532}$>15%) and fluorescence capacity up to $0.7\times10^{-4}$. This mixture is marked
by grey color on aerosol mask in Fig.12b. The pollen index provided by SILAM over Lille on the
midnight, is above 4.0, so the presence of pollen particles is expectable.

The presence of pollen is supported also by the profiles of $\delta_{532}$ and $\delta_{1064}$, shown in Fig.13.

At low heights $\delta_{1064}$ exceeds $\delta_{532}$ and the ratio $\dfrac{\delta_{1064}}{\delta_{532}}$ is about 1.4 at 0.5 km. However, inside the

elevated layer this ratio decreases and becomes about 0.8 at 2.25 km, which indicates that mixture

composition changed. For the same height range, the fluorescence capacity decreases from $0.6\times10^{-4}$

to $0.3\times10^{-4}$ while $A^{\beta}_{532/1064}$ gradually increases from 0.75 to 1.25 which can be due to decrease of

pollen contribution.

As follows from Table 2, in the lower layer the values of $S_{355}$ and $S_{532}$ are close (about $48\pm7$

sr). However, in elevated layer $S_{532}$ increases to $70\pm7$ sr, while $S_{355}$ remains the same. Higher

values of $S_{532}$, in respect to $S_{355}$, are typical for aged smoke (e.g. Müller et al., 2005; Hu et al.,

2022). Moreover, $A^{\beta}_{355/532}$ significantly exceeds $A^{\alpha}_{355/532}$, which was also reported for aged smoke.

Thus, based on intensive properties only, we could classify this layer as "smoke". However, due

to low fluorescence capacity, in our approach we identify it as "urban".

***11 August 2021: contacting layers of smoke and urban aerosol***

Separation of smoke and urban particles is a challenging task for Mie – Raman lidar,

because both types have small effective radius, and similar depolarization ratios $\delta_{532}$. However,

the fluorescence capacity of smoke is about factor 4-5 higher than that of urban aerosol, which

allows their reliable separation. The analyses of the measurements in the night 11-12 August 2021

are shown in Fig.14. The RH decreases with height from 70% to 40% inside 500 m – 2250 m range.

The main part of aerosol is concentrated below 2500 m and two height intervals can be

distinguished. Above approximately 1500 m the layer with high fluorescence capacity (up to

$3.0\times10^{-4}$) is observed, while in the layer below 1500 m, the $G_F$ is low, (below $0.8\times10^{-4}$). HYSPLIT

backward trajectories (not shown) indicate that the air masses at 1800 m heights were transported

from North America, so these may contain wield fire smoke.

On the $\delta_{532}$-$G_F$ diagram (Fig.15a) the single pixel measurements in 500 m – 1400 m and

1400 m – 2500 m height ranges are shown by different colors. The cluster of points, corresponding

to the upper layer, is localized mainly inside the interval $1.8\times10^{-4}<G_F<4.0\times10^{-4}$ and $4\%<\delta_{532}<10\%$,

and can be attributed to smoke. The points corresponding to the lower layer are partly identified

as urban particles, but a part of the points is out of the range and forms a pattern typical for urban

– pollen mixture. The SILAM pollen index in Fig.A1 is above 5.0, so contribution of pollen can

be noticeable. The smoke and urban layers are in contact and the particle mixing occurs, which
increases dispersion within the clusters.
Vertical profiles of $\delta_{532}$ and $A^{\beta}_{532/1064}$ in Fig.16 do not demonstrate significant difference for
upper and lower layers. Meanwhile, the fluorescence capacity increases by factor 4. The lidar ratios
$S_{355}$ and $S_{532}$ in the upper layer, as follows from Table 2, are 45±7 sr and 72±11 sr respectively.
The $A^{\beta}_{355/532}$ significantly exceeds $A^{\alpha}_{355/532}$ (2.2±0.2 and 1.0±0.2 respectively), so based on
intensive parameters, the upper layer can be also identified as smoke.

*1 April 2021: Dust*
Dust layers transported from Africa are regularly observed over North of France. One such
dust episode took place in the night 1-2 April 2021 and the corresponding spatio-temporal
variations of $\beta_{532}$, $\beta_F$, $\delta_{532}$, and $G_F$ are shown in Fig.17. The dust layer, with depolarization ratio
exceeding 30%, and low fluorescence, extends from approximately 1.0 km to 5.0 km height. The
fluorescence capacity varied inside the layer. In the center it was the lowest (about $0.1 \times 10^{-4}$), but
at the bottom of the layer and near the top, $G_F$ increased up to $(0.2 \div 0.3) \times 10^{-4}$. In Fig.18a, ($\delta_{532}$-
$G_F$ diagram), we observe a cluster of particles, which can be identified as dust. There is also a
second small cluster, attributed to urban aerosols. On the distribution of particle types in Fig.18b
the urban aerosol occurs below 800 m after 23:00 UTC.
Fig.19 provides vertical profiles of $\beta_{532}$, $\delta_{532}$, $\delta_{355}$, $\beta_F$, $G_F$ and $A^{\beta}_{355/532}$ . Measurements at
1064 nm were not available for this episode. Depolarization ratios at 355 nm and 532 nm are close
to 30% through the layer, though at heights below 1.5 km there is small enhancement of $\delta_{532}$ up to
34%. The fluorescence capacity is about $0.4 \times 10^{-4}$ at 1.5 km and it decreases with height to $0.1 \times 10^{-4}$
at 2.5 km. However, this decrease is not accompanied by changes in depolarization ratio. The
backscattering Ångstrom exponent $A^{\beta}_{355/532}$ is sensitive to the enhancement of dust absorption in
UV and can be negative (Veselovskii et al., 2020a). For this episode $A^{\beta}_{355/532}$ decreases with height
(together with $G_F$) to -0.3 at 2.5 km. Similar values of $A^{\beta}_{355/532}$ were observed during SHADOW
campaign in Western Sahara (Veselovskii et al., 2020a). Above 3.75 km both $A^{\beta}_{355/532}$ and $G_F$ start
to increase. Hence, dust properties change with height and this change is not revealed on $\delta_{532}$
profile. We should mention, that in publication of Veselovskii et al. (2020a), increase of the dust
imaginary part in UV also did not lead to changes in $\delta_{532}$.

Application of our new "Fluorescence – Depolarization" based approach to six episodes

considered in this section, demonstrates its ability to discriminate several aerosol types. The first
step in validation of the results presented, could be comparison of the particle properties for
obtained aerosol types with corresponding values, used in existing typing algorithms. Table 3
provides the range of variation of particle intensive properties from publications of Burton et al.,
(2013), synthetic values used in NATALI algorithm (Nicolae et al., 2018) and parameters used in
the algorithm of Papagiannopoulos et al. (2018) for the urban, smoke and dust particles. The table
contains also the range of properties variation for the episodes considered in current study for the
same aerosol types.  Parameters chosen in different algorithms, even for the same aerosol type,
vary in a wide range, and the values observed in this study mainly match this range of variation.
We observe higher values of $A^{\beta}_{355/532}$ for urban and smoke particles, and for dust, $A^{\beta}_{355/532}$ could be
negative. Still, the values obtained in this study and the values used by other algorithms are in
reasonable agreement.

**Conclusion**

The results presented in this study can be considered as the first important step in the

combination of Mie – Raman and fluorescence lidar data. In approach presented, only two
intensive parameters are used for classification: the particle depolarization ratio $\delta_{532}$ and the
fluorescence capacity $G_F$. These parameters are chosen because they are specific for different types
of aerosol and can be calculated with high spatio-temporal resolution. Moreover, $\delta_{532}$ and $G_F$ can
be calculated at lower altitudes, compared to extinction related parameters, such as lidar ratio and
extinction Angstrom exponent. Thus classification, in principle, is possible at ranges with
incomplete geometrical overlap. Finally, computation of $\beta_F$ does not demand the use of reference
height, only calibration of relative sensitivity of the channels is needed. Thus, aerosol classification
is possible, even in the presence of low-level clouds.

Though only two aerosol properties are considered, the use of fluorescence provides

advances in aerosol classification. Analysis of numerous observations, performed at Lille
University for the period 2020 – 2021, demonstrates the possibility to separate four types of
aerosols, such as dust, smoke, pollen and urban. Moreover, we are able to identify the layers
containing the liquid water particles and ice. The number of determined aerosol classes can be
increased, by considering the particle mixtures. In particular, "pure" dust can be considered
separately from "polluted" one, which can be discriminated by lower values of the depolarization
ratio.
Fluorescence technique is especially promising for separation of smoke and urban particles,
because fluorescence capacity of smoke is about factor five higher. The important advantage of
fluorescence measurements is the ability to identify the biological particles in the atmosphere, such
as pollen, which are usually not included in the classification schemes, based on Mie-Raman
observations. At the same time, our observations demonstrate that biological particles are
frequently observed during Spring – Autumn seasons and may contribute significantly to the
aerosol composition inside the PBL. The developed approach allows to identify aerosol types with
high spatio-temporal resolutions, which is estimated to be 60 m for height and less than 10 minutes
for time, for the current instrumental configuration. Such resolution provides an opportunity for
investigating the dynamics of aerosol mixing in the troposphere.
The next step in algorithm development will be to include additional particle properties.
We plan to include the backscattering Angstrom exponents and the depolarization spectral ratios
($\delta_{355}/\delta_{532}$ and $\delta_{532}/\delta_{1064}$), which can be also calculated with high spatio-temporal resolutions. The
fluorescence capacity depends on the relative humidity, due to the effects of hygroscopic growth.
Thus, information about spatio-temporal distribution of RH should be included in the analysis. It
is also important to combine our algorithm with existing classification schemes, which we plan to
consider in the near future.

*Data availability*. Lidar measurements are available upon request
(philippe.goloub@univ-lille.fr).
*Author contributions*. IV processed the data and wrote the paper. QH and TP performed the
measurements. PG supervised the project and helped with paper preparation. BB prepared
algorithm for aerosol classification. MK developed software for data processing.
.

*Competing interests*. The authors declare that they have no conflict of interests.

**Acknowledgement**
We acknowledge funding from the CaPPA project funded by the ANR through the PIA under
contract ANR-11-LABX-0005-01, the "Hauts de France" Regional Council (project CLIMIBIO)
and the European Regional Development Fund (FEDER). ESA/QA4EO program is greatly
acknowledged for support of observation activity at LOA as well as OBS4CLIM Equipex project
funded by ANR. Development of algorithm for aerosol typing was supported by Russian Science
Foundation (project 21-17-00114). The SILAM model is acknowledged for providing pollen
simulations.

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

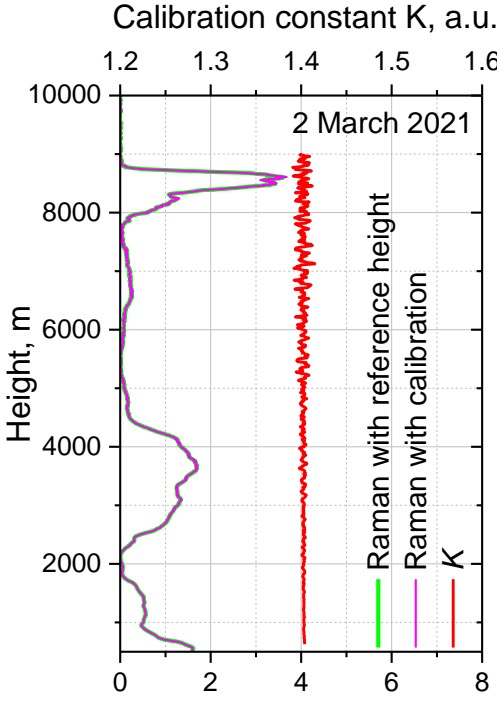

Fig.1. Backscattering coefficients at 532 nm for period 19:00 – 20:00 UTC on 2 March 2021
calculated from Mie-Raman observations using the reference height as Ansmann et al. (1992)
(green) or the calibration constant as in Eq 5. (magenta). The profile of calibration constant $K$ is
shown with red line.



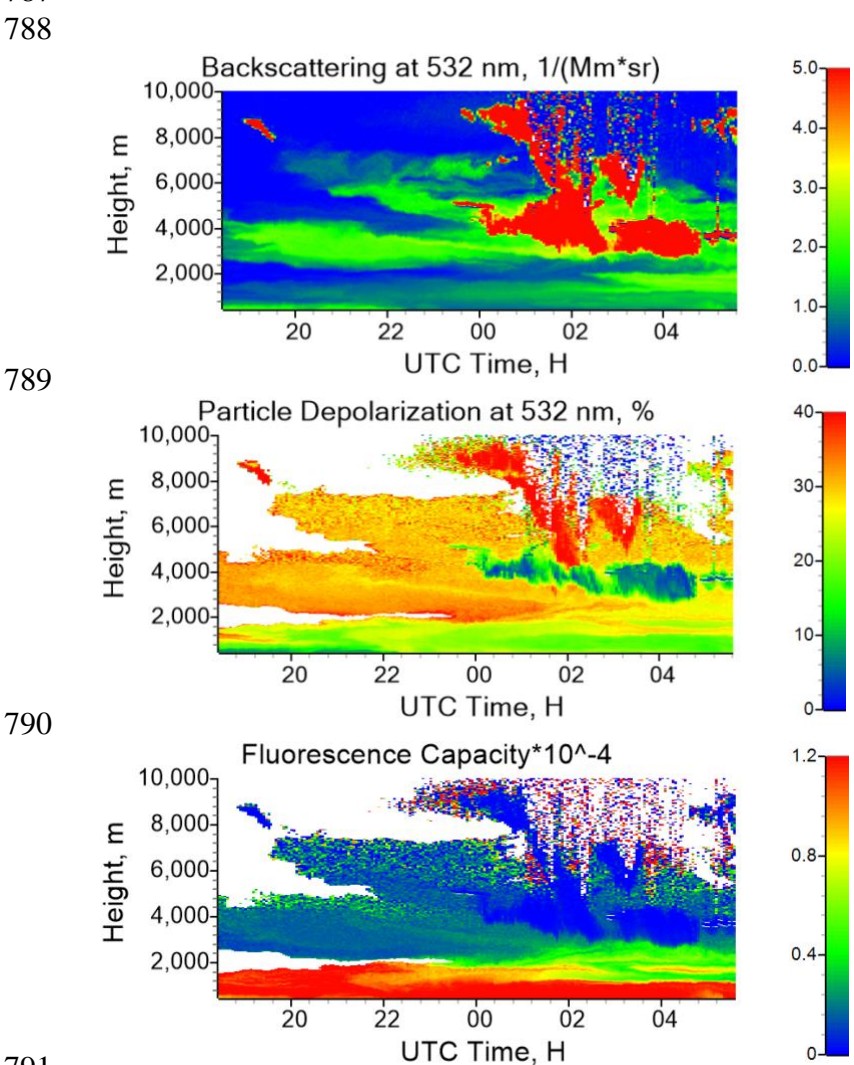



Fig.2. Spatio-temporal distributions of the backscattering coefficient $\beta_{532}$, the particle
depolarization ratio $\delta_{532}$ and the fluorescence capacity $G_F$ in the night 2-3 March 2021. The
backscattering coefficient $\beta_{532}$ is calculated with the modified Raman method. The values of $\delta_{532}$,
and $G_F$ are shown for $\beta_{532} > 0.2$ Mm$^{-1}$sr$^{-1}$.


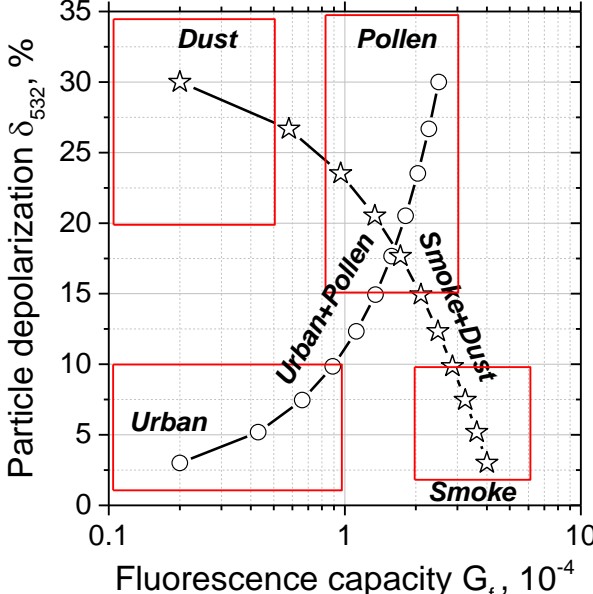

Fig.3. Aerosol typing with $\delta_{532}$-$G_F$ diagram. The ranges of the particle parameters variation for
dust, pollen, smoke and urban aerosol are given by rectangles. The symbols show the results of
simulation performed for pollen+urban (circles) and smoke + dust (stars) mixtures. Relative
contribution of pollen (smoke) to the total backscattering $\beta_{532}$ varied in $0-1.0$ range with step 0.1.
Particle parameters used in calculations are given in the text.

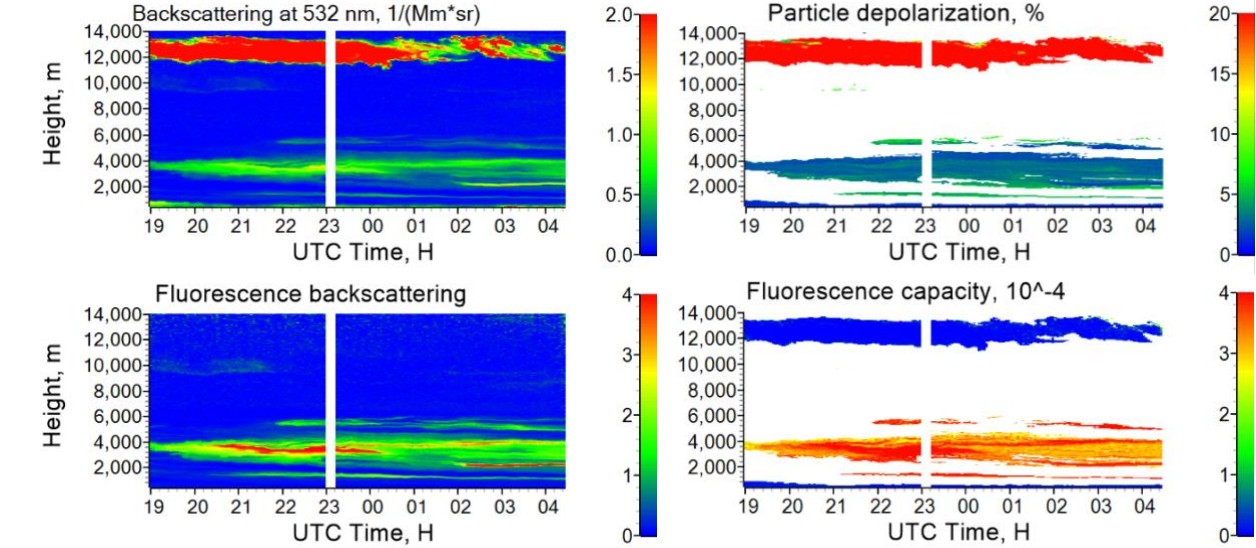

Fig.4. Spatio-temporal distributions of the backscattering coefficient $\beta_{532}$, the fluorescence
backscattering coefficient $\beta_F$ (in $10^{-4}$ Mm$^{-1}$sr$^{-1}$), the particle depolarization ratio $\delta_{532}$; and the
fluorescence capacity $G_F$ in the night 12-13 September 2020. Calculation of $\delta_{532}$ and $G_F$ was not
performed for $\beta_{532}<0.2$ Mm$^{-1}$sr$^{-1}$. The values of backscattering coefficient and depolarization ratio
of ice clouds are high (above 20 Mm$^{-1}$sr$^{-1}$ and 40% respectively) and are off scale on the maps
presented.

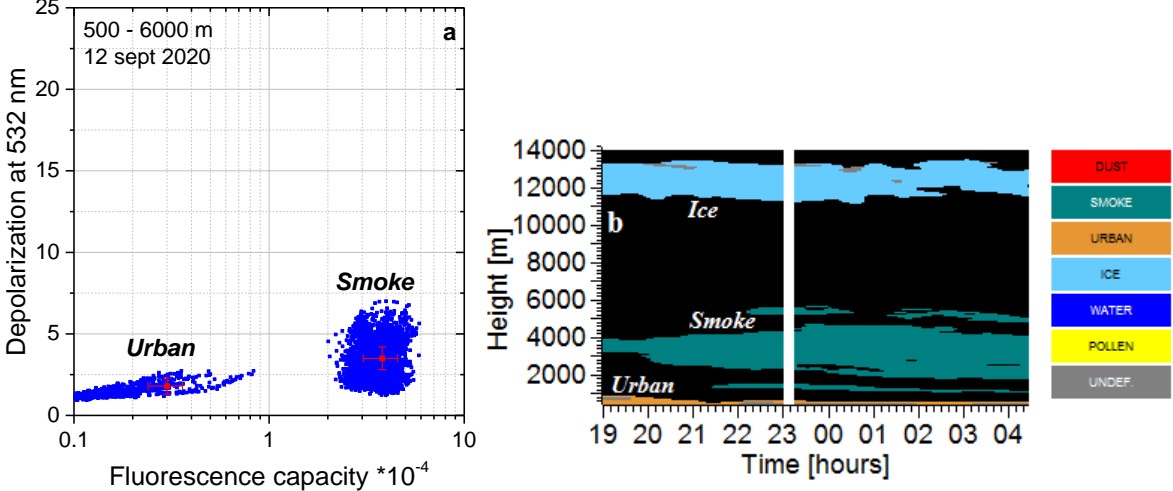

Fig.5 (a) The $\delta_{532}$-$G_F$ diagram for data from Fig.4 in 500 – 6000 m height range, red crosses show
the uncertainty of the measurements. (b) Spatio-temporal distribution of aerosol types in the night
12-13 September 2020. Grey color shows undefined aerosol type, while measurements with
$\beta_{532}<0.2$ Mm$^{-1}$sr$^{-1}$ are marked by black color.



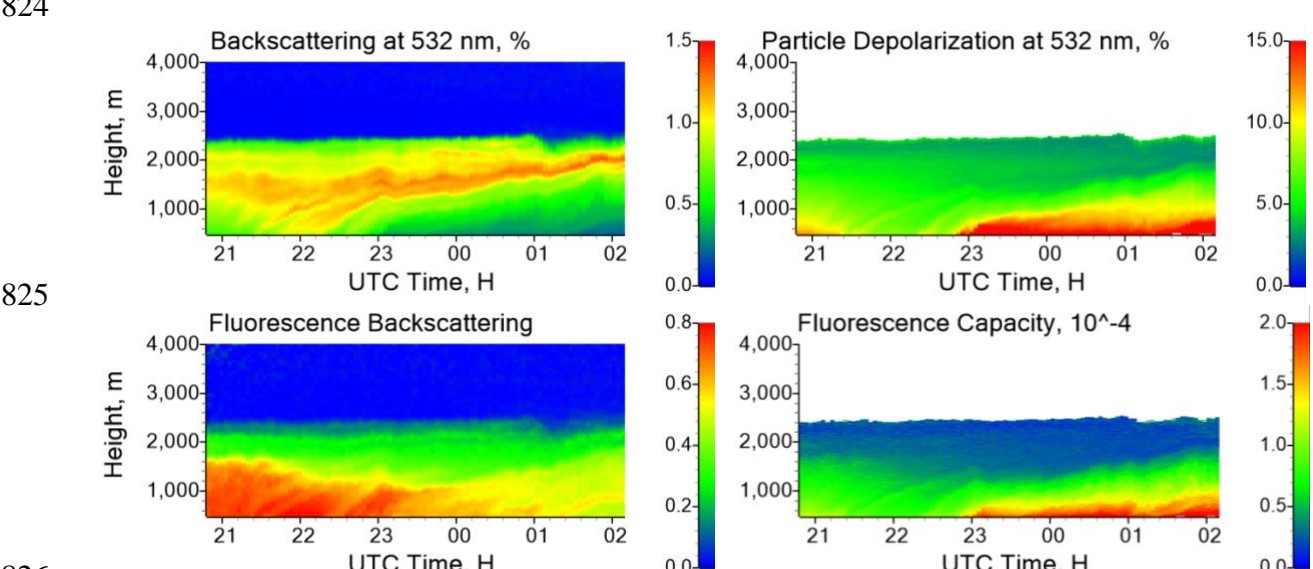


Fig.6. Spatio-temporal distributions of the backscattering coefficient $\beta_{532}$; the fluorescence
backscattering coefficient $\beta_F$ (in $10^{-4}$ Mm$^{-1}$sr$^{-1}$); the particle depolarization ratio $\delta_{532}$; and the
fluorescence capacity $G_F$ in the night 30-31 May 2020. The values of $\delta_{532,}$ and $G_F$ are shown for
$\beta_{532}>0.2$ Mm$^{-1}$sr$^{-1}$.

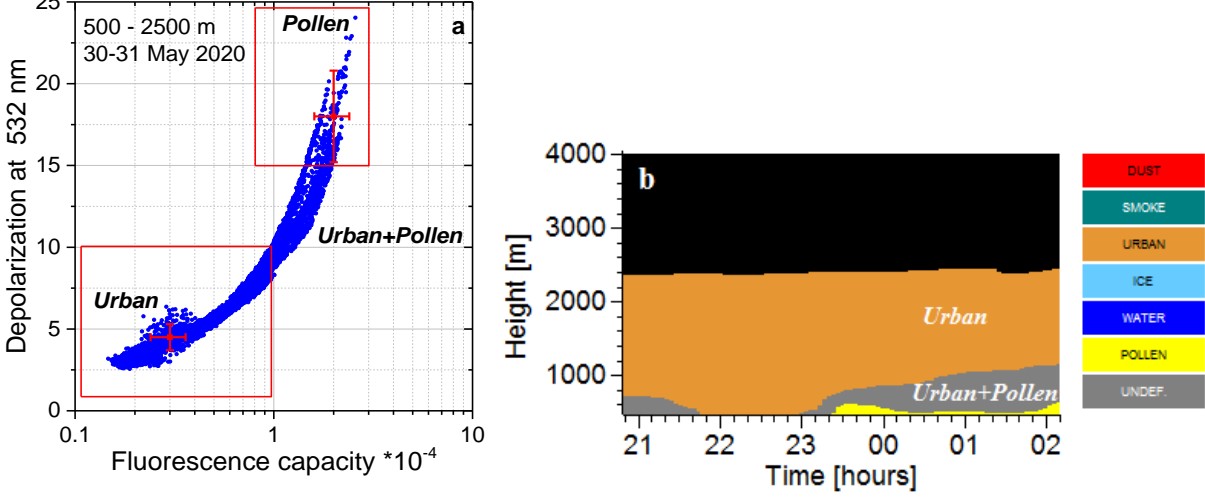

Fig.7. (a) The $\delta_{532}$-$G_F$ diagram for observations in 500 m – 2500 m height range and (b) spatio-
temporal distribution of aerosol types on the night 30-31 May 2020. Grey color shows undefined
aerosol type, which is a mixture of urban and pollen for this case. Measurements with $\beta_{532}<0.2$
Mm$^{-1}$sr$^{-1}$ are marked by black color.


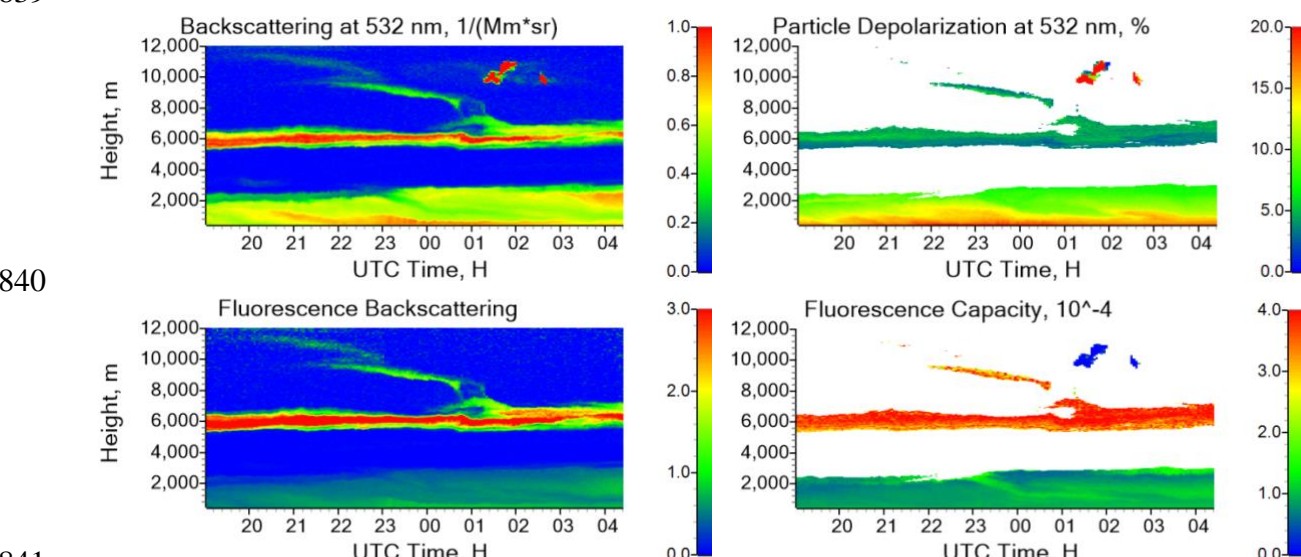



Fig.8. Spatio-temporal distributions of the backscattering coefficient $\beta_{532}$, the fluorescence
backscattering coefficient $\beta_F$ (in $10^{-4}$ Mm$^{-1}$sr$^{-1}$), the particle depolarization ratio $\delta_{532}$, and the
fluorescence capacity $G_F$ in the night 14 – 15 September 2020. The values of $\delta_{532,}$ and $G_F$ are
shown for $\beta_{532}$>0.2 Mm$^{-1}$sr$^{-1}$.

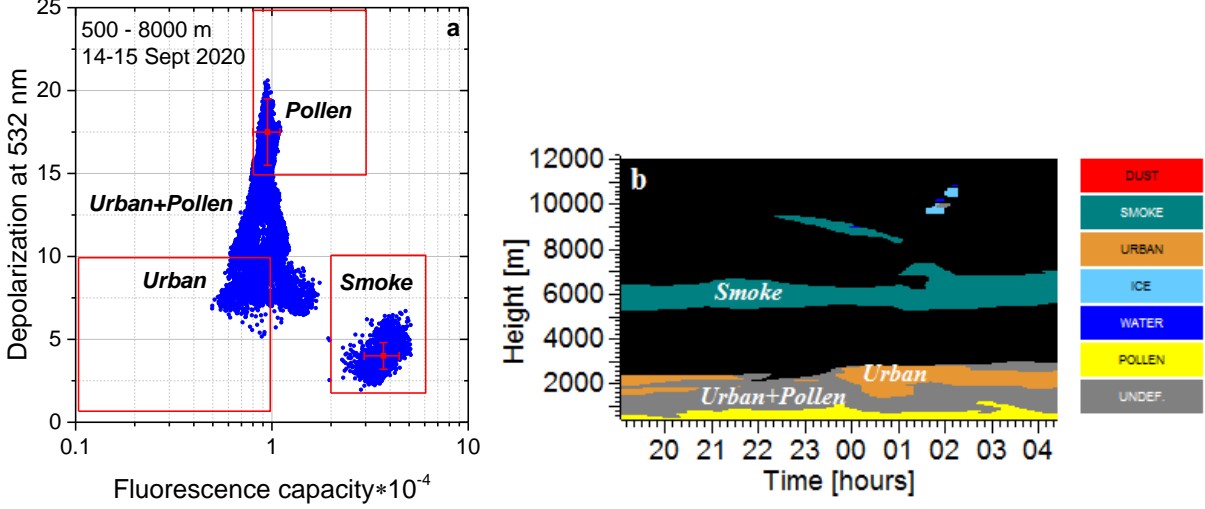


Fig.9. (a) The $\delta_{532}$-$G_F$ diagram for observations in 500 m – 8000 m height range and (b) spatio-
temporal distribution of aerosol types in the night 14 – 15 September 2020. Grey color shows
undefined aerosol type, which is a mixture of pollen, urban and smoke particles. Measurements
with $\beta_{532}$<0.2 Mm$^{-1}$sr$^{-1}$ are marked by black color.


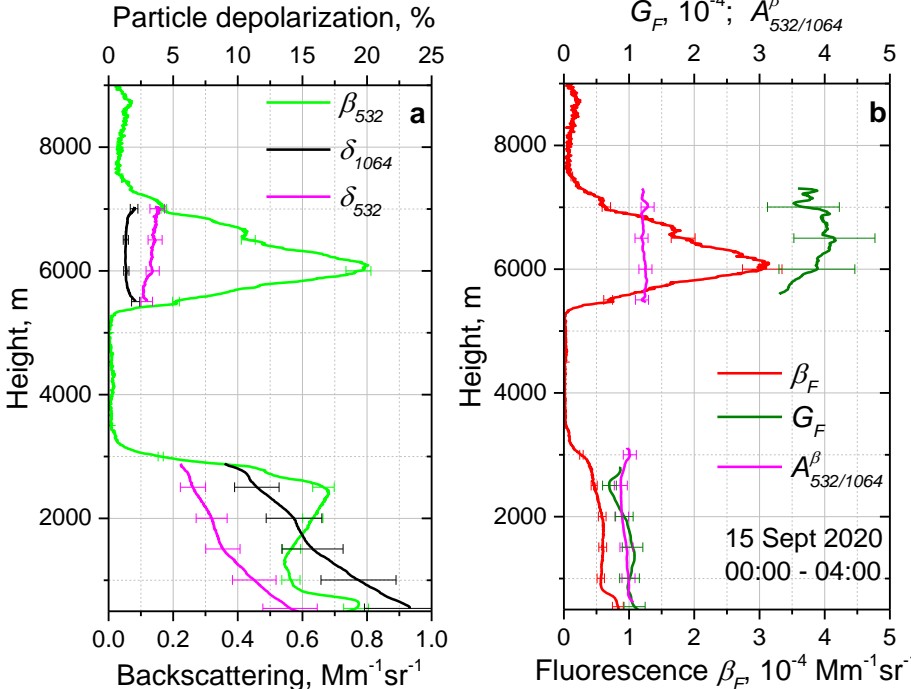


Fig.10. Vertical profiles of (a) backscattering coefficient $\beta_{532}$ and particle depolarization ratios $\delta_{532}$,
$\delta_{1064}$; (b) fluorescence backscattering $\beta_F$, fluorescence capacity $G_F$ and backscattering Angstrom
exponent $A^{\beta}_{532/1064}$ on 15 September 2020 for period 00:00 – 04:00 UTC.



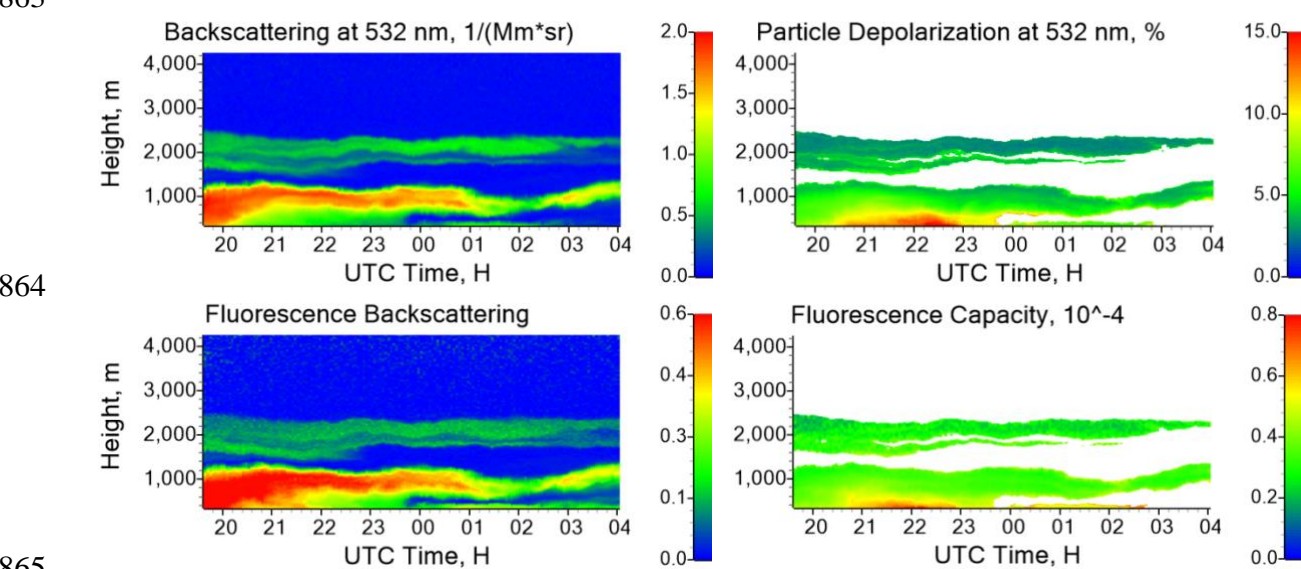


Fig.11. Spatio-temporal distributions of the backscattering coefficient $\beta_{532}$, the fluorescence
backscattering coefficient $\beta_F$ (in $10^{-4}$ Mm$^{-1}$sr$^{-1}$), the particle depolarization ratio $\delta_{532}$; and the
fluorescence capacity $G_F$ in the night 10 – 11 April 2020. Measurements are performed at an angle
of 45 dg to horizon. The values of $\delta_{532}$, and $G_F$ are shown for $\beta_{532}>0.2$ Mm$^{-1}$sr$^{-1}$.

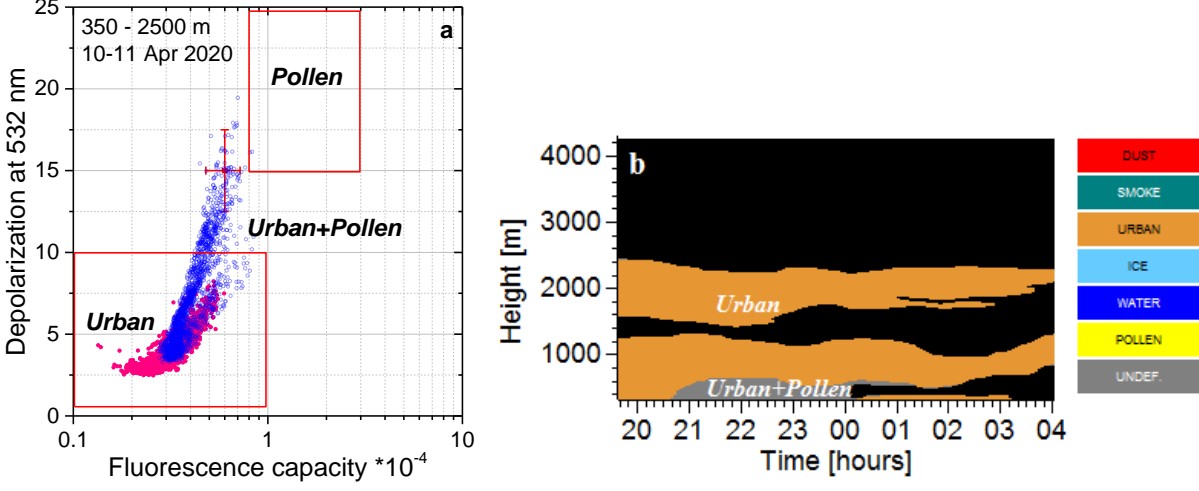

Fig.12. (a) The $\delta_{532}$-$G_F$ diagram for observations in 350 – 1500 m (blue symbols) and 1500 – 2500
m (pink symbols) height ranges. (b) Spatio-temporal distribution of aerosol types in the night 10
– 11 April 2020. Grey color shows undefined aerosol type, which is a mixture of urban and pollen
for this case. Measurements with $\beta_{532}<0.2$ Mm$^{-1}$sr$^{-1}$ are marked by black color.

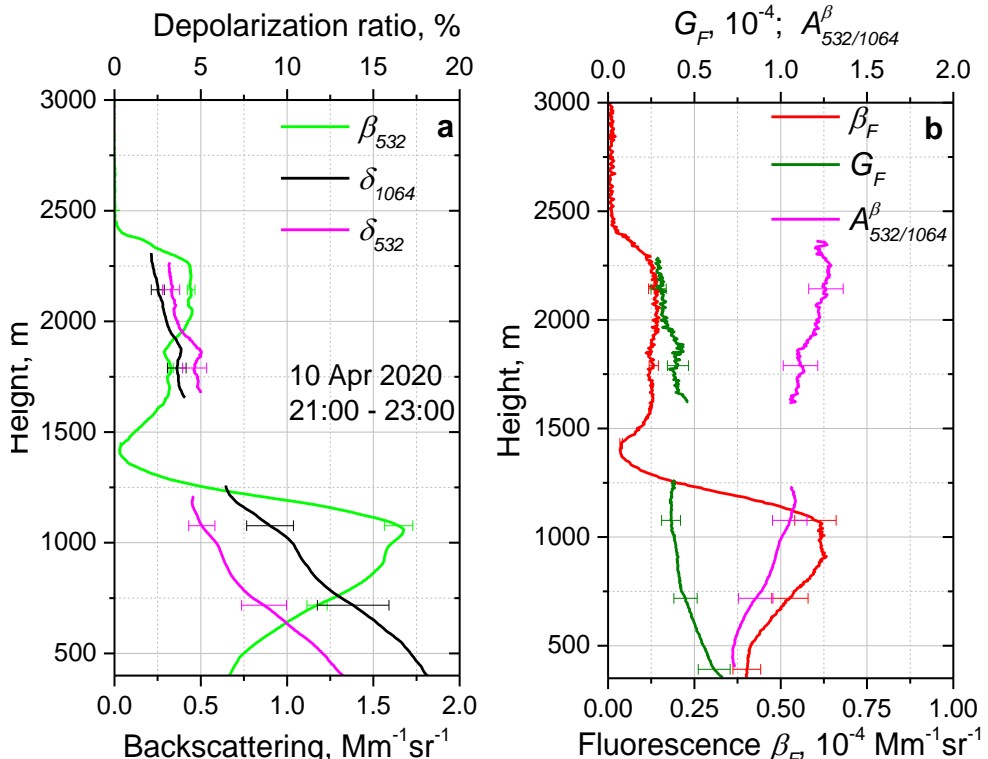


Fig.13. Vertical profiles of (a) backscattering coefficient $\beta_{532}$ and particle depolarization ratios $\delta_{532}$,
$\delta_{1064}$; (b) fluorescence backscattering $\beta_F$, fluorescence capacity $G_F$ and backscattering Angstrom
exponent $A^{\beta}_{532/1064}$ on 10 April 2020 for period 21:00 – 23:00 UTC.


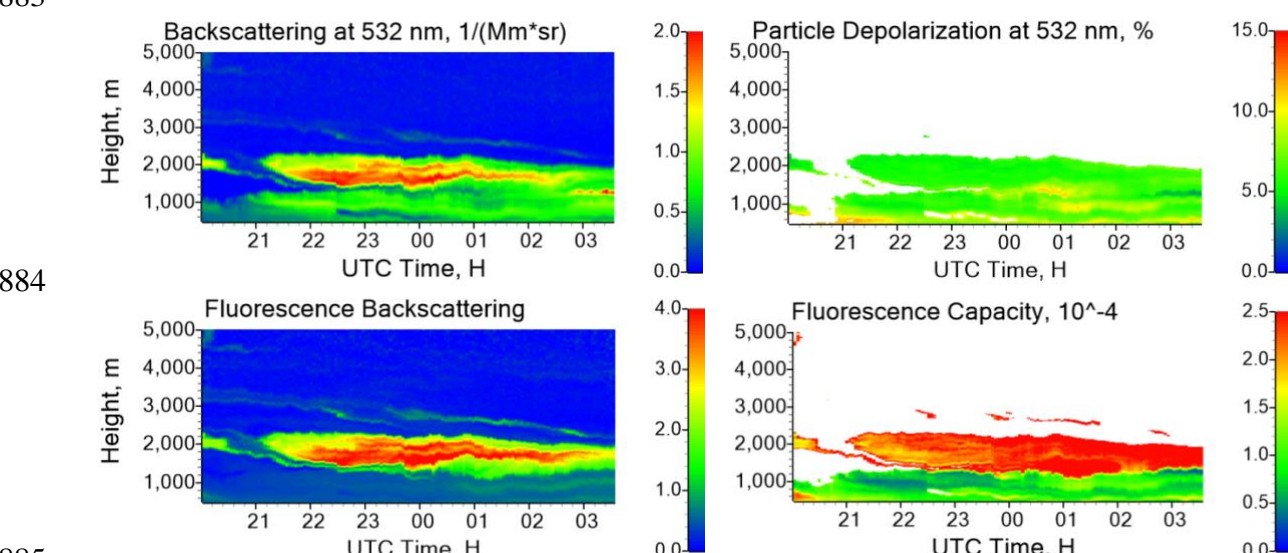



Fig.14. Spatio-temporal distributions of the backscattering coefficient $\beta_{532}$, the fluorescence
backscattering coefficient $\beta_F$ (in $10^{-4}$ $Mm^{-1}sr^{-1}$), the particle depolarization ratio $\delta_{532}$, and the
fluorescence capacity $G_F$ in the night 11 – 12 August 2021. The values of $\delta_{532}$, and $G_F$ are shown
for $\beta_{532}>0.3$ $Mm^{-1}sr^{-1}$.

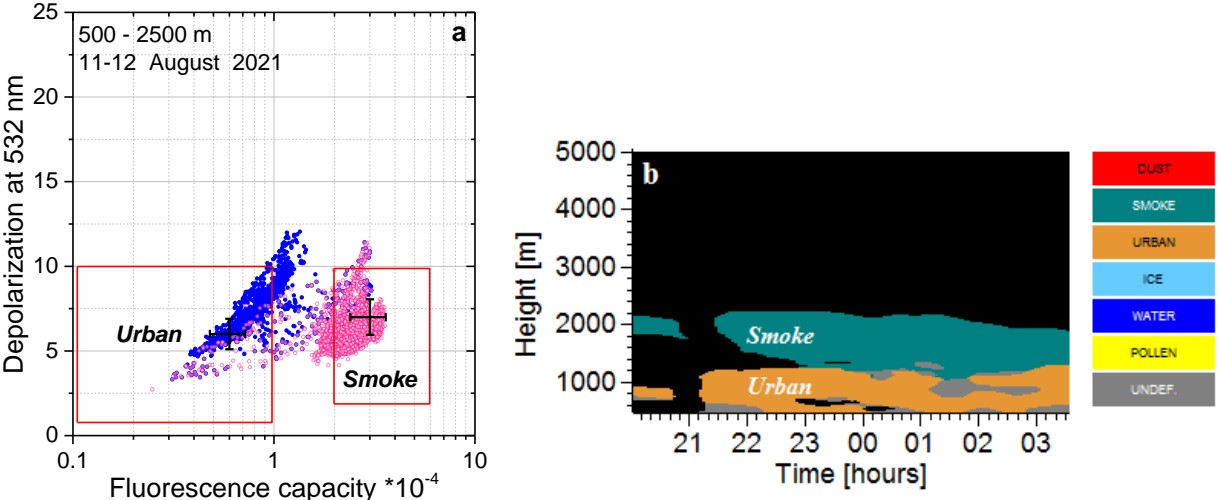


Fig.15. (a) The $\delta_{532}$-$G_F$ diagram for observations in 500 – 1400 m (blue symbols) and 1400 – 2500
m (pink symbols) height ranges. (b) Spatio-temporal distribution of aerosol types in the night 11-
12 August 2021. Measurements with $\beta_{532}<0.3$ $Mm^{-1}sr^{-1}$ are marked by black color.




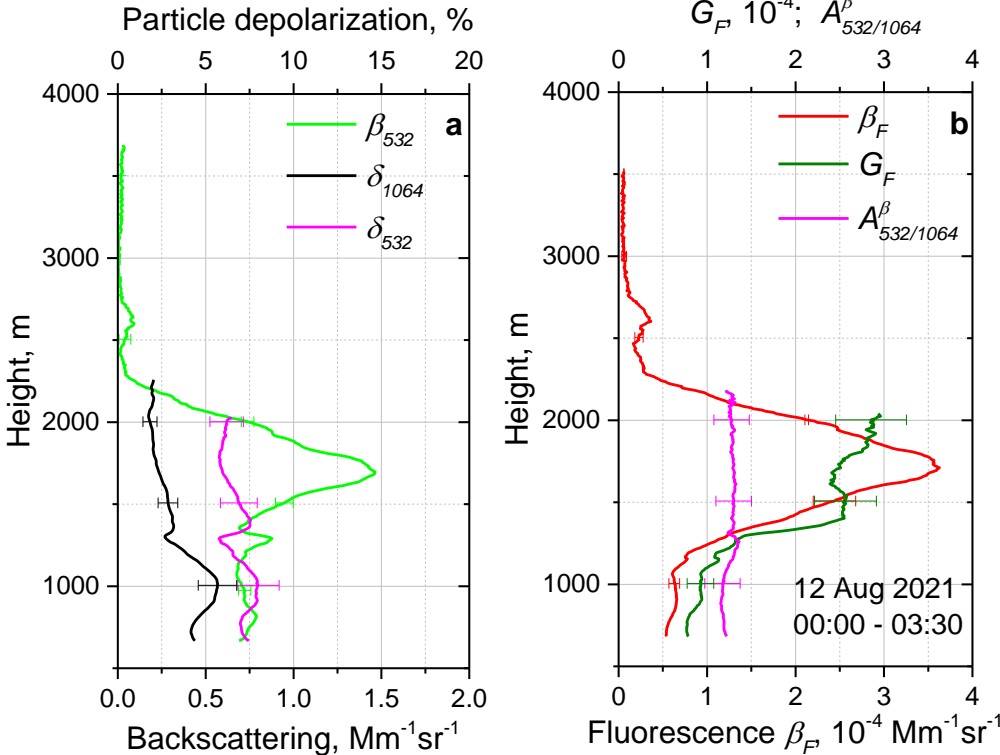

Fig.16. Vertical profiles of (a) backscattering coefficient $\beta_{532}$ and particle depolarization ratios $\delta_{532}$,
$\delta_{1064}$; (b) fluorescence backscattering $\beta_F$, fluorescence capacity $G_F$ and backscattering Angstrom
exponent $A^{\beta}_{532/1064}$ on 12 August 2021 for period 00:00 – 03:30 UTC.


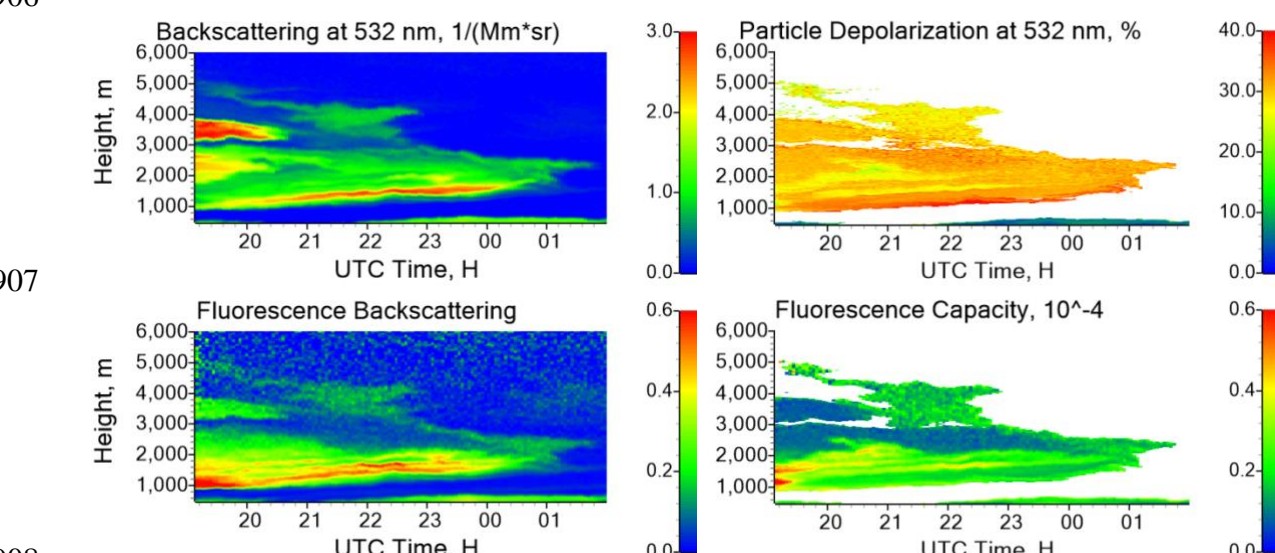


Fig.17. Height – temporal distributions of the backscattering coefficient at 532 nm $\beta_{532}$, the
fluorescence backscattering coefficient $\beta_F$ (in $10^{-4}$ $Mm^{-1}sr^{-1}$), the particle depolarization ratio at
532 nm $\delta_{532}$, and the fluorescence capacity $G_F$ in the night 1-2 April 2021. The values of $\delta_{532,}$ and
$G_F$ are shown for $\beta_{532}>0.3$ $Mm^{-1}sr^{-1}$.

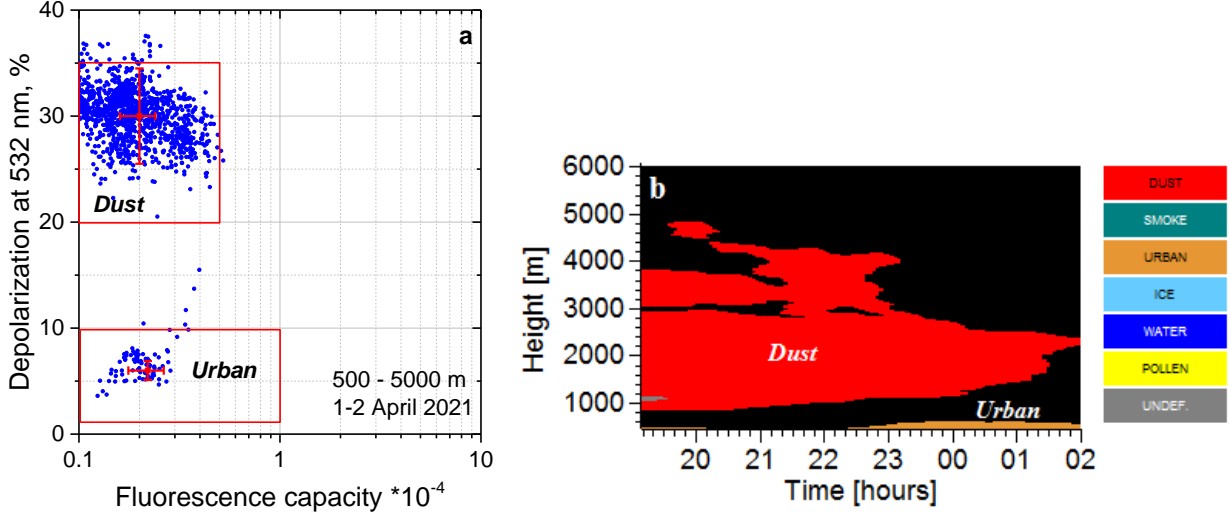

Fig.18. (a) The $\delta_{532}$-$G_F$ diagram for observations in 500 – 5000 m height range and (b) spatio-
temporal distribution of aerosol types in the night 1-2 April 2021. Measurements with $\beta_{532}<0.3$
$Mm^{-1}sr^{-1}$ are marked by black color.

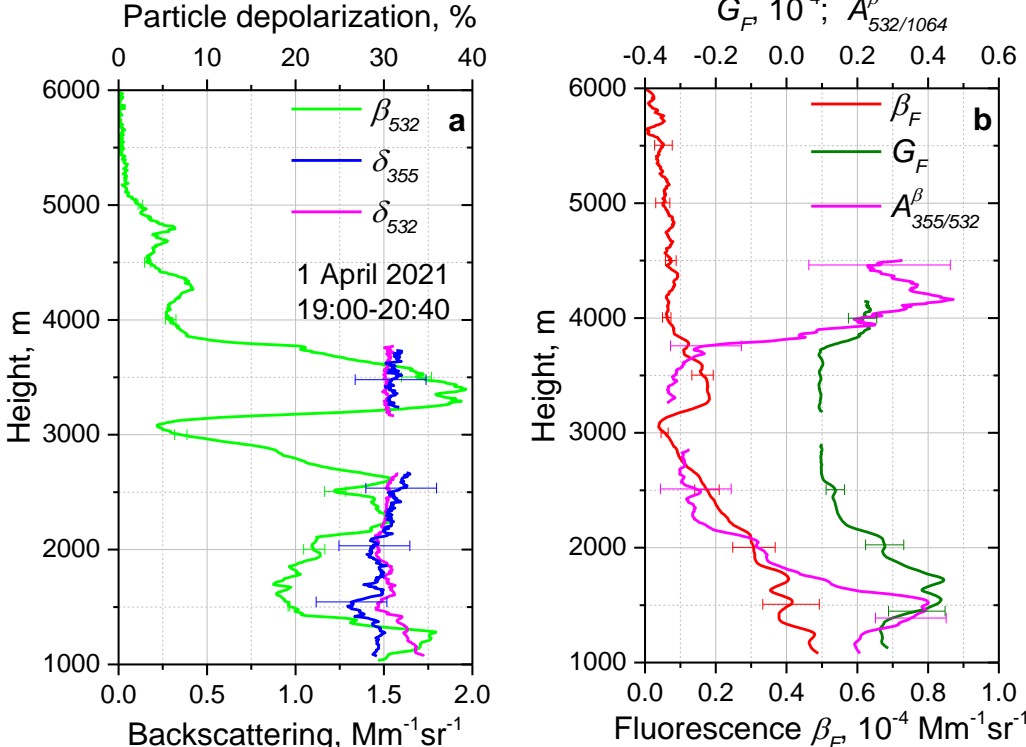

Fig.19. Vertical profiles of (a) backscattering coefficient $\beta_{532}$ and particle depolarization ratios $\delta_{532}$, $\delta_{355}$; (b) fluorescence backscattering $\beta_F$, fluorescence capacity $G_F$ and backscattering Angstrom exponent $A^{\beta}_{355/532}$ on 1 April 2021 for period 19:00 – 20:40 UTC.

Table 1. Ranges of particle depolarization $\delta_{532}$ and fluorescence capacity $G_F$, which were used for aerosol classification.

| Aerosol type | $\delta_{532}$, % | $G_F$, (×10$^{-4}$) |
|---|---|---|
| Dust | 20 - 35 | 0.1 – 0.5 |
| Pollen | 15 - 35 | 0.8 – 3.0 |
| Urban | 1 - 10 | 0.1 – 1.0 |
| Smoke | 2- 10 | 2.0 – 6.0 |
| Ice | >40 | <0.01 |
| Water | <5 | <0.01 |




Table 2. Intensive particle parameters such as the lidar ratios ($S_{355}$, $S_{532}$), particle depolarization ratios ($\delta_{355}$, $\delta_{532}$, $\delta_{1064}$), extinction
($A^{\alpha}_{355/532}$) and backscattering ($A^{\beta}_{355/532}$, $A^{\beta}_{532/1064}$) Angstrom exponents for six episodes, analyzed in this work. Parameters are given for
chosen height – temporal intervals and the types of aerosol are determined from fluorescence measurements.

| Date | Time, UTC | H, km | Type | $S_{355}$, sr | $S_{532}$, sr | $\delta_{355}$, % | $\delta_{532}$, % | $\delta_{1064}$, % | $A^{\alpha}_{355/532}$ | $A^{\beta}_{355/532}$ | $A^{\beta}_{532/1064}$ |
|---|---|---|---|---|---|---|---|---|---|---|---|
| 10.04.2020 | 21:00-23:00 | 0.9-1.1 | Urb.+Poll. | 48±7 | 48±7 | 5.0±1.0 | 6.0±1.0 | 10±1.5 | 1.3±0.2 | 1.4±0.2 | 1.0±0.2 |
|  |  | 2.0-2.2 | Urban | 50±7 | 70±10 | 7.0±1.0 | 3.5±0.7 | 3.0±0.6 | 1.1±0.2 | 2.0±0.2 | 1.2±0.2 |
| 30.05.2020 | 21:00-02:00 | 1.8-2.0 | Urban | 60±9 | 55±8 | 3.6±0.8 | 4.0±0.8 | 5.7±1.0 | 2.0±0.2 | 1.6±0.2 | 1.2±0.2 |
| 12.09.2020 | 20:00-23:00 | 3.2-3.8 | Smoke | 50±7 | 80±12 | 4.5±1.0 | 3.0±0.6 | 2.0±0.4 | 1.0±0.2 | 2.2±0.2 | 1.2±0.2 |
| 15.09.2020 | 00:00-04:00 | 1.4-1.6 | Pollen | 40±6 | 37±6 | 9.5±1.5 | 8.0±1.5 | 15±2.5 | 1.6±0.2 | 1.4±0.2 | 0.9±0.2 |
|  |  | 5.8-6.2 | Smoke | 45±7 | 70±10 | 9.0±1.5 | 3.5±0.7 | 1.4±0.3 | 0.8±0.2 | 2.0±0.2 | 1.2±0.2 |
| 01.04.2021 | 19:00-20:40 | 2.25-2.5 | Dust | 57±8 | 52±8 | 30±4.5 | 30±4.5 | - | 0±0.2 | -0.3±0.2 | - |
| 11.08.2021 | 22:00-24:00 | 1.0-1.2 | Urban | 42±7 | 55±8 | - | 8.0±1.2 | 5.7±0.8 | 1.3±0.2 | 1.5±0.2 | 1.1±0.2 |
|  |  | 1.5-2.0 | Smoke | 45±7 | 72±11 | - | 6.0±0.9 | 2.5±0.5 | 1.0±0.2 | 2.2±0.2 | 1.2±0.2 |




Table 3. Intensive particle parameters from publications of Burton et al., (2013); Nicolae et al.,
(2018); and Papagiannopoulos et al., (2018) together with values observed in current study for the
urban, smoke and dust particles.

| | Burton et al., 2013 | Nicolae et al., 2018 | Papagiannopoulos et al., 2018 | This study |
|---|---|---|---|---|
| | Urban | Continental (rural) | Clear continental | Urban |
| $S_{355}$, sr | | 43-54 | 50±8 | 42 - 60 |
| $S_{532}$, sr | 43-81 | 52-53 | 41±6 | 55 -70 |
| $A^{\alpha}_{355/532}$ | - | 1.2-1.3 | 1.7±0.6 | 1.1 -2.0 |
| $A^{\beta}_{355/532}$ | - | 1.0-1.6 | 1.3±0.3 | 1.5 - 2.0 |
| $A^{\beta}_{532/1064}$ | 0.49-1.3 | 0.54 – 1.0 | 1.0±0.3 | 1.1 - 1.2 |
| Smoke | | | | |
| $S_{355}$, sr | - | 56-72 | 81±16 | 40 - 50 |
| $S_{532}$, sr | 46-87 | 81-92 | 78±11 | 70 - 80 |
| $A^{\alpha}_{355/532}$ | - | 1.1-1.3 | 1.3±0.3 | 0.8 - 1.0 |
| $A^{\beta}_{355/532}$ | - | 1.4 -2.1 | 1.2±0.3 | 2.0 - 2.4 |
| $A^{\beta}_{532/1064}$ | 0.48-1.6 | 0.7-0.8 | 1.3±0.1 | 1.2 |
| Dust | | | | |
| $S_{355}$, sr | - | 43-46 | 58±12 | 57 |
| $S_{532}$, sr | 41-57 | 44-49 | 55±7 | 52 |
| $A^{\alpha}_{355/532}$ | - | 0.88-0.92 | 0.3±0.4 | 0 |
| $A^{\beta}_{355/532}$ | - | 0.91-0.97 | 0.3±0.2 | -0.3 |
| $A^{\beta}_{532/1064}$ | 0.49-0.68 | 0.16-0.22 | 0.4±0.1 | - |


**Appendix.** Pollen index provided by SILAM

The  SILAM  is a  chemical  transport  model, developed  by  the Finnish  Meteorological
Institute (Sofiev et al., 2015). It provides information on atmospheric composition, air quality, and
pollen. In the pollen module of SILAM, six pollen types (alder, birch, grass, mugwort, olive,
ragweed) are considered. The pollen index is defined as a quantitative measure of the severity of
the pollen season and a proxy of the allergenic exposure (Sofiev et al., 2012, 2017). This higher
the pollen index is, the more pollen grains in the atmosphere and the higher allergy risk.  Fig. A1
shows the maps of pollen index in 4 cases. According to the description of SILAM model, the
pollen index is labeled as "very high", when its value is greater than 4.0.

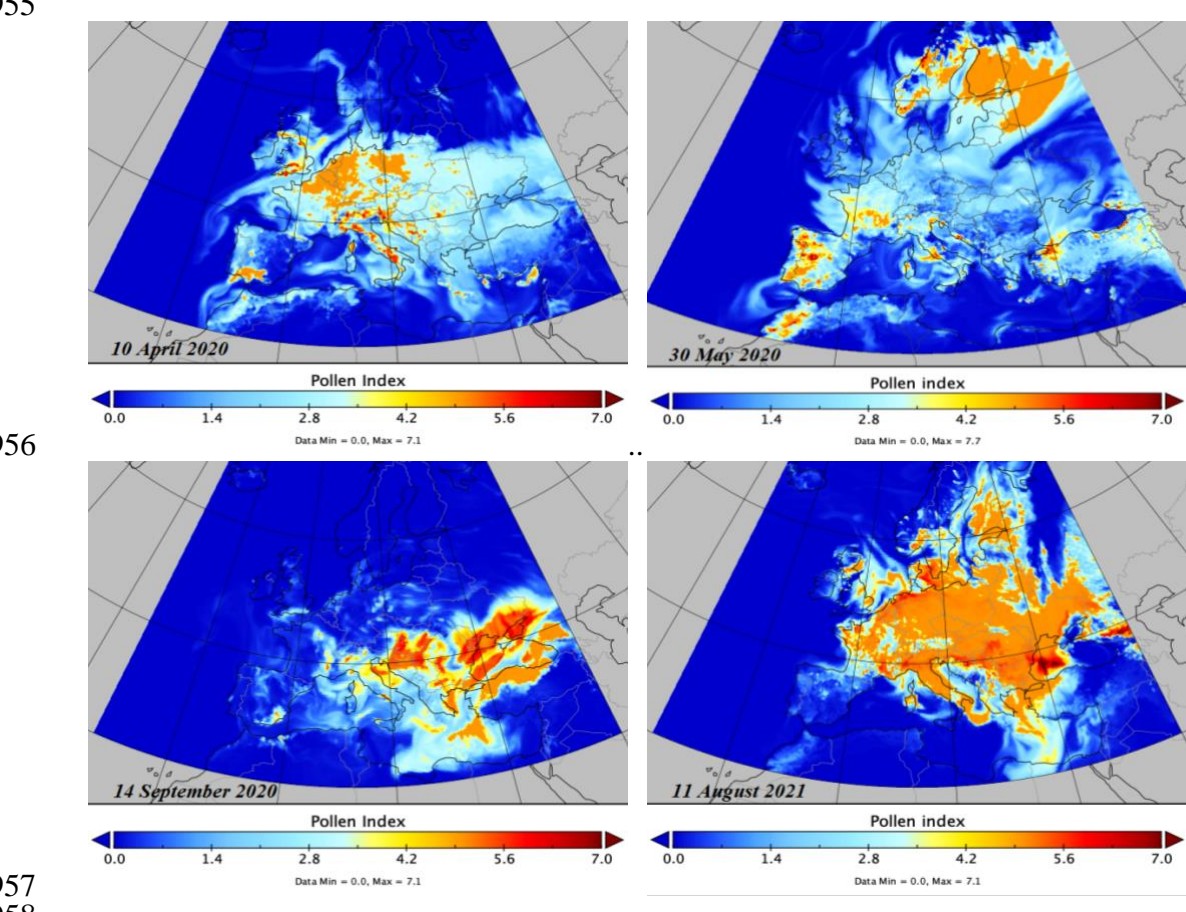



Fig.A1. Pollen index provided by SILAM for 10 April 2020, 30 May 2020, 14 September 2020
and 11 August 2021. The levels of pollen index are – very low (<1.0), low (1.0÷2.0), moderate
(2.0÷3.0), high (3.0÷4.0) and very high (>4.0).

