# Peer review of "Combining of Mie-Raman and fluorescence observations: a step forward in aerosol"

_Atmospheric Measurement Techniques, 2022_

## Author Response (AR1)

First of all we would like to thank the reviewer for very detailed comments and useful suggestions, which helped us to improve the revised manuscript

*The manuscript describes several case studies of lidar observations where fluorescence observations combined with lidar depolarization shows significantly different properties for pollen, smoke, dust and anthropogenic aerosol. I'm excited to see the potential of these new measurements, which give completely independent and orthogonal information about aerosol particles, at single bin resolutions, significantly increasing the information available for aerosol typing. The case studies are a nice selection of different types and mixtures and interesting to see.*

*The manuscript seems to suffer from an identity problem, however. Mostly it is an illustrative set of cases studies that demonstrate differences in the two-dimensional space of fluorescence capacity and particle depolarization. It includes nice analysis of some mixtures of types as well. However, the paper claims to be an algorithm description paper, and for that purpose, analysis of a few hand-selected case studies really isn't sufficient, and the mixture analysis doesn't exactly fit, because it is not part of the algorithm. Apparently in consequence of this uncertainty about the desired focus of the paper, some aspects of the paper seem superficial, or rather, inconsistent in depth. The inferences in the paper about the types seem very reasonable, but many are not backed up by any independent information or compared with other methods of classification, which should be done to demonstrate the validity of the new algorithm, particularly if this is the algorithm description paper. Also there's insufficient information about how the thresholds in the algorithm were chosen. In the analysis of the case studies, there should be a consistent effort to include complimentary information to validate the case identifications using other measurements (in situ or other lidar measurements that reveal type) and backtrajectories. And if a major focus of the paper is to showcase the performance of a new (and better) classification algorithm, then the results should be shown on a bulk of data in addition to the case studies, and comparisons with other classification methods should be made and discussed.*

The goal of this manuscript is to demonstrate that the fluorescence – depolarization diagram allows to separate different types of aerosol and provides new independent information on aerosol type, which can be used in classification schemes. The reviewer is right, at current stage of research it is not appropriate to call it "algorithm", so we escape this term in the revised manuscript.

In the revised manuscript we tried to follow the reviewer recommendations. We added a table, containing the particle intensive parameters for the cases considered (lidar ratios at 355 and 532 nm; depolarization ratios at 355, 532 and 1064 nm; and the backscattering and extinction Angstrom exponents). Another table provides the range of variation of particle intensive properties from different typing algorithms for the urban, smoke and dust particles. The table contains also the range of parameters variation for episodes from current study for the same aerosol types.

The back-trajectory analysis is included.

In Appendix we added four maps with SILAM pollen index, for the episodes where the presence of the pollen was revealed. We hope, that all this improved the manuscript.

Specific comments:

*L24. "and their mixtures". The mixture analysis is an interesting part of the paper, and apparently new compared to the authors' other papers, but it appears it's not really part of the classification algorithm, in the sense that mixture analysis can only be done on a case-by-case basis. Any discussion about that? This could be clarified in the abstract. Also, the mixture analysis is not even mentioned in the introduction. Discussing it there would help to clarify the novel aspects of the paper.*

The mixture analysis is an important but in the manuscript presented we just identify the main mixture components, based on the patterns in depolarization – fluorescence diagram. Quantification of the mixture composition is the next step in our research and corresponding algorithm is in preparation at the moment. We removed from Abstract the mentioning of mixture analysis.

*L73-75. I very much agree that adding independent aerosol information will improve classification, but this specific point is unconvincing. Yes, the variables used for classification so far have variability within types but there's nothing to suggest that this won't also be true for fluorescence capacity, is there? So, I'm not sure this is exactly the right motivation.*

The advantage of fluorescence is strong variation of fluorescence capacity between some aerosol types. For example, $G_F$ of smoke can up to one order higher, comparing to urban aerosol, allowing to separate these particles. So we think, that synergy of existing algorithms with fluorescence measurements should improve identification. Another important advantage is that $G_F$ and depolarization can be derived with high spatio – temporal resolution, so almost single pixel typing becomes possible.

*L105. Good point that the resolution is higher since fluorescence capacity can be calculated using data at a single bin, unlike extinction or other quantities related to extinction. This seems particularly useful for Raman measurements.*

Yes.

*L105-107. Veselovskii et al. 2021a is referenced extensively in the introduction, including to say that it already demonstrates the ability of the 2-d measurement space to separate all the aerosol types. I couldn't follow how the purpose and scope of this paper is different from 2021a.*

In that paper we just formulated the idea and plotted averaged data for several observations on the depolarization – fluorescence diagram. In this manuscript we evaluate the aerosol type mask with almost singe pixel resolution. Corresponding paragraph is added to the revised manuscript.

*L183-193. Calculation of the backscatter coefficient using a calibration constant sounds so straightforward, that I didn't realize that it hadn't been done before. This is great. It's good to see a relatively straightforward innovation discovered and put into practice that will produce a significant amount of additional retrievals, in profiles when the reference height is not accessible to the lidar.*

We are very pleased, that Reviewer liked our approach

*L231-232. Add an earlier reference for spectral dependence of the depolarization ratio, Burton et al. 2015.*
Added

*L240-241. Since line 223 just said that Veselovskii et al. 2021a already demonstrated that the two dimensional diagram can separate types, is the part about mixtures the main purpose of this manuscript? If so, the abstract and intro should make that clearer and the examples should be chosen to align with that purpose.*

We modified Introduction, to show that the main goal is to provide aerosol type mask with high spatio-temporal resolution. The patterns at $\delta_{532}$-$G_F$ diagram help to identify the mixture, but at current stage we can not characterize it quantitatively.

*L248-249. Burton et al. (2012) or Burton et al. (2013), referenced elsewhere in the manuscript, is an earlier lidar aerosol classification methodology with depolarization ratio ranges listed for common types.* Added

*L247. "The ranges are based on results obtained in LOA". The algorithm is a simple thresholding method in two dimensions, so the ranges are the single most important aspect of the algorithm description. This statement is much too vague to support and explain how the ranges were derived, and I'm eager to know more. What results? From cases published in other publications? From a completely independent subset of cases than the results shown in this manuscript? Are the results only inferences from the lidar measurements of depolarization and fluorescence capacity, or do they include other coincident measurements that provide stronger evidence for the type identifications? Is there a set of training cases that are classified using other external measurements and/or source information? Are the cases shown in this paper the training cases or are they independent cases that demonstrate the validation of the algorithm? All this should be part of the methodology discussion.*

We agree with reviewer and completely modified that section. We added:

**Dust**. The depolarization ratio $\delta_{532}$ of Saharan dust near the source regions is up to 35% (Veselovskii et al., 2020a), but after transportation and mixing with local aerosol $\delta_{532}$ can be as low as 20% (Rittmeister et al., 2017). In many studies, the dust with decreased depolarization ratio is classified as "polluted dust" (e.g. Burton et al., 2012, 2013). At a moment, we do not introduce the discrimination between the two subtypes and mark as "dust" the particles with $20\%<\delta_{532}<35\%$, and $0.1\times10^{-4}<G_F<0.5\times10^{-4}$.

**Smoke**. In 2021-2022 we regular observed over Lille the smoke layers originated from Californian and Canadian forest fires (Hu et al., 2021). The particle depolarization and fluorescence capacity of transported smoke changed from episode to episode and for classification we choose the ranges $2\%<\delta_{532}<10\%$, $2\times10^{-4}<G_F<6\times10^{-4}$. At this stage we do not discriminate "fresh" and "aged" smoke, and the range of $\delta_{532}$ variation is similar to the one, used in classification of Burton et al. (2012).

**Pollen**. The pollen over north of France is usually mixed with other aerosols, and the particles, which we mark as "pollen" are actually the mixtures. Depolarization ratio of clean pollen varies strongly for different taxa. For birch pollen, Cao et al. (2010) reported $\delta_{532}=33\%$, and in the measurements over Finland during birch pollination (Bohlmann et al., 2019), observed values of $\delta_{532}$ up to 26%. The observations over Lille during pollen season (Veselovskii et al., 2021a) rarely revealed values $\delta_{532}$ exceeding 20%. Based on that observations, we type as "pollen" the particles mixtures with $15\%<\delta_{532}<30\%$, and $0.8\times10^{-4}<G_F<3.0\times10^{-4}$.

**Urban**. This type of aerosol includes a variety of particle types (e.g. sulfates, soot) and its parameters may depend on the relative humidity. Based on our measurements inside the boundary layer, for classification we choose the ranges $1\%<\delta_{532}<8\%$, and $0.1\times10^{-4}<G_F<0.8\times10^{-}$

[4]. Similar range for $\delta_{532}$ is used in classification of Burton et al. (2012). Urban and smoke particles both have a low depolarization, but the fluorescence capacity of smoke is almost one order higher, so these particles can be reliably discriminated.

**Ice and water clouds**. Both types of the clouds have low fluorescence capacity $G_F$ $<0.01\times10^{-4}$. However, the ice clouds are usually observed at the heights, where fluorescence signal is low and can not be used for classification. Thus above ~8 km the ice cloud are identified by high depolarization ratio $\delta_{532}>40\%$. Depolarization ratio of the liquid water clouds is usually affected by the effects of the multiple scattering, so for their identification we use $\delta_{532}<5\%$."

*Figure 3. The mixing lines all go through the box that's marked "pollen". This highlights the unavoidable weakness of typing with just two dimensions. Presumably, anything that falls within this box needs context to distinguish between pollen, a pollen mixture, or a smoke-dust mixture that has nothing to do with pollen. Identification by context (particularly where supporting measurements are available) is fine for the purpose of case studies, but there must be significant potential for misidentification in the automated algorithm, I suppose. It would be good to discuss weaknesses as well as strengths of the approach.*

Yes, aerosols are always the mixtures. So this problem is attributed not only to the presented, but also to all existing classification algorithms. Next step in our research is the increase of the number of parameters used and quantifications of mixture components.
It is true, that dust – smoke mixture, considered just at one point at depolarization – fluorescence diagram can be recognized as pollen. This is why it is important to consider all the data obtained during the session. We tried to show in this manuscript that the single pixel data for different mixtures provide different patterns, as shown in Fig.3. In our analysis we always observed this kind of patterns, and it helps to get idea about mixture composition.

*L268. Clouds are also shown in the aerosol typing masks and line 308 mentions both ice and water droplets, so the thresholds values for ice and water droplets should also be included in Table 1.*
*Figs 4,5. It's confusing that the ice cloud is only partially included in this example. It's shown in the type mask, but not discussed, and it's not shown in the scatter plot in Fig 5a. It's included in Fig 4, but apparently off-scale. The authors should decide whether they want to include the cloud in their analysis and discussion or not. If not, cut off the plots at an altitude below the cloud. If so, rescale Figure 4, include it in Fig 5 and add discussion about cloud.*

The parameters for ice and water particles are added to the Table 1. The ice clouds, however, are normally observed at high altitudes, where fluorescence signal is very weak, so corresponding points at depolarization – fluorescence diagram demonstrate strong scattering. Usually we identified the ice crystals from depolarization measurements only, and this is why we don't show them in Fig.5a. Corresponding comment is added to the revised manuscript.

*Figure 5 and similar figures. What's the purpose of the boxes and cross-hairs in the fluorescence vs. depolarization diagrams? The boxes would probably be more useful to readers if they were all the same, and used the values from Table 1. That way, we can see visually how the identified types fall into the broad category already established. I can guess that the crosshairs represent the mean and (probably) standard deviation of identified pure types, but those aren't discussed anywhere in the paper.*

In our revised manuscript the boxes correspond to Table 1. The crosses show uncertainty of our measurements, due to statistical errors and uncertainty of calibration. Corresponding comment is added to revised manuscript.

*L315-321. The explanation of the smoothing procedure is missing something. Z is a number, but the classification IDs are not numbers that can be added and weighted, but just labels. How are the classifications convolved with Z? Just guessing, I suppose the fluorescence capacity and depolarization ratio are what's averaged using the Z-weightings, and then the classification is done on these smoothed measurements instead? Please clarify in the text.*

To make it more clear, we modified corresponding section in the revised manuscript and extended description.
Briefly:
We construct several 'raw' matrices with dimensions equal to primary data matrices (one matrix for each aerosol type (dust, pollen, etc)). If at the first stage some single pixel data point (i,j) is classified as, e.g., pollen, the corresponding value in the 'pollen' matrix is set to 1, otherwise it is set to 0. Then each of these matrices is separately convoluted with the Gauss kernel Z. And, after the convolution, the values for each pixel data (i,j) are being compared. If, e.g., the 'dust' matrix (after the convolution) contains maximal value at the point (i,j) among all the matrices (after the convolution), then the point (i,j) is finally classified as 'dust'.

*L339 and 341 and elsewhere. I'd suggest avoiding describing values as "typical" and expand the description to be more specific. For instance, perhaps this is within the ranges seen in your previous publications and/or other publications for cases that have been identified as smoke and urban based on independent data? "Typical" is a bit dangerous, in that it implies a generality that is not established after only a few handfuls of case studies, particularly since the case study identifications seem to mostly be rather dependent on expectations about the typical values. Statements like this unfortunately seem to be quoted and referenced repeatedly so that they become ingrained without becoming better supported. After all, we now know that it is quite common for smoke (in the upper troposphere and stratosphere) to have depolarization values that are much larger than this, and previously published ranges of depolarization for urban aerosol also include significantly larger depolarization values than this.*

Agree. We tried to follow this recommendation in revised manuscript
It is true, that aged smoke depolarization ratio at 532 nm in stratosphere can be as high as ~20%. We should mention also, that at 1064 nm the depolarization ratio of smoke in our measurements (even in upper troposphere) never exceeded 5%. This is one more reason to include this depolarization ratio in typing scheme at next stage.

*L347. Says that the fluorescence capacity can decrease as a function of relative humidity, explaining a range of variables. Why does it produce variability rather than reducing the fluorescence capacity uniformly?*
The water uptake increases the particle backscattering, but does not change the fluorescence. As a result, the fluorescence capacity decreases. The RH, changes with height, which can lead to increase of single pixel data scattering inside the cluster.

*L361-367 and Figure 6-7. I agree that the shape of the curve in Figure 7a is very striking and reminiscent of a mixing line. However, I also just read in the previous section that fluorescence capacity is strongly impacted by relative humidity, making me wonder quantitatively how much impact RH has, compared to the impact of mixing. Is there a model (theoretical or empirical) of G_F dependence on relative humidity? The RH profile should be added to Figure 8 (and all the other profile figures). Another aspect that puzzles*

*and surprises me is the increased G_F specifically in parts of the curtain where the backscatter is lower. This hints that the variation in G_F might be quite strongly related to RH; alternately that the pollen is more diffuse and widespread than the urban aerosol, which I think would be unusual. A curtain of RH (perhaps from MERRA-2 since there is insufficient sonde data to produce a curtain) and/or backtrajectories might help make the scenario more clear.*

Unfortunately, we had no collocated RH measurements. The sonde measurements in UK show that RH increased from 40% to 70% with height. The value of the fluorescence capacity changed for one order of magnitude, and such strong change in $G_F$ can not be explained by the particle hygroscopic growth. For example, from the recent publication of Sicard et al., increase of $\beta_{532}$ in this RH range for urban aerosol is below factor 1.5. (Sicard, M., Fortunato dos Santos Oliveira, D. C., Muñoz-Porcar, C., Gil-Díaz, C., Comerón, A., Rodríguez-Gómez, A., and Dios Otín, F.: Measurement Report: Spectral and statistical analysis of aerosol hygroscopic growth from multi-wavelength lidar measurements in Barcelona, Spain, Atmos. Chem. Phys. 22, 7681–7697, 2022). Corresponding comment is added to revised manuscript.

The hygroscopic growth can contribute to the backscattering near the PBL top. However, at low altitudes RH is about 40%, so increase of $G_F$ is probably due to decrease of urban particles contribution to the total backscattering (thus pollen contribution becomes more visible). We tried to use MERRA-2 data, but at low altitudes the modeled parameters differed strongly from observations.

*L368-369. It's good that 1064 nm depolarization is included here, because in general, the more data shown, the better the patterns can be understood. However, the text highlights larger values of 1064 nm depolarization to support the inference of pollen, but that's also true for urban aerosol (e.g. Burton et al. 2012). Then "both depolarization ratios decrease with height" as the pollen concentration decreases (L372), but 1064 continues to be larger than 532, so again this is not definitive. Any further comment about this?*

Yes, urban aerosol may also have $\delta_{1064}$ exceeding $\delta_{532}$. But absolute values of depolarization for pollen are significantly higher. So when at low altitudes we observe high $G_F$, and high depolarization, the observed $\delta_{1064} > \delta_{532}$ corroborates presence of pollen.

*This case and the first case were also included in earlier publications by the same authors. The papers make different analyses of them, so that's fine, but does this mean they also contributed information relevant to producing the ranges used in the algorithm? If so, they are not such good examples to illustrate the performance of the typing algorithm.*

The typing is performed on a base of $G_F$-$\delta_{532}$ measurements only. We used these examples, because the aerosol origin was analyzed in our previous publications. Besides, measurements on 30 May 2020 demonstrate very characteristic pattern for urban – pollen mixture.

The vertical profiles of particle parameters for 30 May were presented in our recent paper, so we decided to exclude Fig.8 from revised manuscript. We just provide the reference.

*Figure 8 L 715. Why were the profiles created for 21:00-23:00 instead of a later time where the curtain shows pollen at lower altitudes and mixing is discussed? Is this a mistake?*

Sorry, this was mistake.

*L376-377. I'm not finding the explanation for the lack of variability in the backscatter angstrom exponent to be very convincing. It appears to be saying that the urban particles are growing due to humidification exactly in balance with the effective dry particle size decreasing due to less pollen? (if so, this needs support).*
*Perhaps some quantitative modeling would help. How small of a backscatter Angstrom exponent would be expected for high concentration of pollen, and just how much contribution to the backscatter is there (based on the mixing model) and how much change in Angstrom would you therefore expect? What confuses me is that the fluorescence capacity also mixes linearly according to the backscatter partition, so if there was really too little backscatter contribution to be noticeable, wouldn't that also mean there would be little variation in G_F as well?*

In revised manuscript, this section was completely modified. We agree with reviewer, that behavior of backscatter Angstrom 532/1064 is puzzling. However, the observation presented, could be strongly influenced by hygroscopic growth, which decreases both depolarization and the fluorescence capacity. The backscattering Angstrom exponent strongly (and in complicated way) depends on refractive index, particle size and particle shape. The modeling of the BAE for different mixture compositions is important, but it is out of scope of this research. Just want to mention, that that in publication of Bohlmann et al. (2019) the BAE (at depolarization ratio ~20%) is about 1.0. Which is quite high value and pollen content over Finland is significantly higher than over Lille. So this aspect needs additional research and additional measurements during strong pollen episodes.

*L392-393. Unfortunately, the SILAM website only provides current forecast data, so please make the relevant data available as a supplement or shown in a figure. Also, what kind of pollen was it?*

In situ measurements at the roof of the building demonstrate presence of significant amount of grass pollen. We added to the revised manuscript (as Appendix) the SILAM maps for four episodes, when presence of pollen was assumed.

L418-419. I'm not quite clear on what the author's intent is here. Is this saying that the algorithm misclassified a mixture as pure urban, or that the mixture only occurs where the classification puts it, but that the two urban layers have quite a lot of difference between them? It would be very helpful (in this case and others) to mark the points in the scatterplots according to the classification result or altitude. I would like to see exactly where the two layers classified as "urban" fall on the apparent mixing line. I think it's interesting that the two layers marked urban have different spectral dependence of depolarization. Backtrajectories would be helpful for this case too, to help understand why the two layers of urban aerosol might have different properties.

To make presentation more clear, we significantly modified this section. First of all, in depolarization – fluorescence diagram in Fig.12 we show the points related to the upper and lower layers by different colors. Back trajectories analysis shows that air masses in both layers are transported from England. So this is probably pollution. Points related to the upper layer are inside the range for 'urban' aerosol. Points in the lower layer, are partly outside of this range, so the aerosol type is undefined. We assume that this is the mixture of urban and pollen particles, because we have particles with high depolarization and fluorescence capacity (still not high enough to be classified as "pollen"). This mixture is marked by grey color and it is located below 750 m. The maps with SILAM pollen index are added to the revised manuscript as Appendix. On the midnight of 10-11 April 2020 the pollen loading is modeled by SILAM as moderate. Thus yes, properties of layers are different. Upper layer is urban, while in lower layer below 1 km the urban particles are mixed with pollen.

L420. "typical for urban-pollen mixture". Actually the mixing curve is significantly to the left of the curve in Figure 3, suggesting that the pure pollen in this mixture is not "typical" compared to the ranges given in the table, but is more of an edge case with relatively low fluorescence capacity.

Yes, fluorescence capacity is lower than usual, so this not pure pollen. We added corresponding comment to the text.

L430-436.  This is a very nice case to demonstrate contrast in fluorescence between different types.  But the type identification is entirely made by inference using the two classification dimensions without any other support such as in situ measurements, backtrajectories, or other lidar-measured quantities like 1064 nm depolarization, lidar ratio or angstrom exponents. It's great that two measurements used for the classification appear to give the ability to make these separations, but for such a key demonstration I think the case studies need to be very well supported.  In general I suggest bolstering the verification of the identifications for all the cases (not just this one) by including all relevant data.  I mean specifically, first of all, other lidar quantities that have been used in previous classification methodologies, including especially lidar ratios, and also 1064 nm depolarization and angstrom exponents for all cases.  Also include RH, backtrajectories and any coincident in situ measurements (especially pollen) for all cases.

In the revised manuscript we added the Table 2, with main intensive particle parameters for all episode considered. The section is modified: we added backtrajectories and analysis of the intensive particle parameters. We have added also Table 3, which compares our observed intensive parameters for dust, smoke, urban with parameters used in existing typing algorithms.

L445 and Figs 15 and 16. The suggested mixing between layers doesn't look convincing.  On the fluorescence vs. depolarization diagram, these intermediate points don't follow a nice mixing line like the other mixing cases, and the boundaries in the measurement curtains appear quite crisp. Could these points be artifacts of the smoothing instead?

Yes, at high gradients of backscattering, smoothing sometimes can provide oscillation. We reprocessed this case with decreased smoothing. The threshold value of $\beta_{532}$ was increased up to 0.3 Mm-1sr-1. Now it is better.

Fig 15. The depolarization especially and perhaps also the fluorescence capacity (outside of the smoke plume) seems to be anti-correlated with backscatter, including in regions that seem unlikely to be pollen-dominated (such as the minimum between the smoke and urban layers). Particulate depolarization is especially susceptible to systematic error, particularly overestimation, at low values of backscatter (Freudenthaler et al. 2009, Burton et al. 2015).  Have you done a systematic uncertainty calculation? (Also this is another case where color coding of the scatterplot by altitude would be useful).

Yes, calculation of depolarization at low $\beta_{532}$ can lead to enhanced uncertainty, especially when high gradients of $\beta_{532}$ present. In reprocessed data we increased threshold value of $\beta_{532}$ up to 0.3 Mm-1sr-1. Oscillations decreased. The same is true for fluorescence capacity.
We estimate uncertainty of our depolarization calibration to be below 15%.

Fig 15-18. Include the data for depolarization and angstrom exponent (and RH) for these cases also.

In the revised manuscript we have added Figures 16, 19 with vertical profiles for these episodes.

L455 "G_F increased ... probably due to the mixing with local pollution".  Does this make sense?  Nothing prior to this in the manuscript suggests that urban pollution has significant fluorescence capacity.  Also, on the scatter plot on Figure 18, there's no suggestion that the higher values of G_F in the dust cluster are correlated with depolarization in any way; that is, they are not following any mixing line. What evidence is there that this is not simply normal variability within dust?  Table 1 shows dust can have G_F up to 0.5.  Why not 0.6?  Also, could some of this variability be correlated with RH?

Dust may have very low fluorescence capacity (0.1*10^-4), while urban particles for some episodes had $G_F$ of 0.8*10^-4, or even higher. Thus mixing of dust with pollutions, in principle, can increase the capacity. But reviewer is right, for case presented, the depolarization ratio did not change significantly with height, while capacity strongly decreased in the center of the layer. It can be variation of dust composition (and so the absorption) through the layer. Unfortunately, at this stage we can not make definite conclusion. Corresponding section is strongly modified in revised manuscript.

For the available dust episodes the fluorescence capacity was mainly below 0.5*10^-4. This is why we used it in Table 1. We may reconsider this range, when more data will be available.

Normally properties of dust are not very sensitive to RH. Increase of RH can only decrease the capacity. However at a moment we don't have collocated RH measurements, so unable to make quantitative conclusions about RH influence.

L485. "during Spring-Autumn seasons". It would be helpful to show a timeseries demonstrating that the pollen signature (elevated depolarization and fluorescence capacity) does NOT occur in winter.

We agree, that this would be useful, but it is beyond the scope of this manuscript. Seasonal variation of aerosol composition over Lille will be the topic of separate research.

Typographical or wording:
L19. What is meant by "single" in "first single version of the algorithm". I suggest delete "single" or reword.
Corrected

L18 and L24. Change particle's to particle.
Corrected

L92. Be specific about which wavelength here.
Done

L247. Define LOA.
Done

L270-281. There should be some discussion or at least references to other analyses of mixtures of aerosols that derive similar equations (especially Eq. 7), e.g. Sugimoto and Lee 2006, Gross et al. 2011, Gasteiger et al. 2011, Tesche et al. 2009, Burton et al. 2014.

The references are added. Derivation of Eq.7 looks very straightforward, so probably no explanations are needed.

L280. Eq. 8. It probably would be good to remind the reader that fluorescence capacity and backscatter in this equation refer to particular wavelengths.
Done

L282. "We assume". I think this is meant to refer only to the demonstration in Figure 3, not a general assertion. If true, perhaps swap the first two sentences of the paragraph to make it less likely to be misread. As mentioned in the introduction, the quantities have a lot of variability even within types, so assuming single values wouldn't be well supported.

Done

L300. "the height resolution is 7.5 m". Is that really the resolution or only the grid spacing? That is, taking the detectors into account, are measurements at adjacent vertical grid points independent?

Yes, this is bin resolution of our detection electronics, and in many cases this resolution was used to calculate the particle properties. However, for elevated layers the fluorescence signal was splined. For typing, the Gaussian smoothing procedure was used. Thus ultimate resolution was about 60 m for height and less than 10 minutes for time.

L342. spell out FBC
The section was modified

L344. Add a reference to the reminder. (I think it is Veselovskii 2020?)
Done

L445. Typo in "0.2-0.3"
Corrected

L663-664. It would be helpful to add "using the reference height as Ansmann et al. 1992 (green) or the calibration constant as in Eq 5. (magenta)". (I read figure captions before the text, so having a bit more detail in the captions is very helpful)
Done

L708. Please add clarification to the caption whether the scatter plot shows data for the entire time period shown in the curtain or only the subset that's included in the profile plots of Figure 8.
This is for entire time period. Added to caption.

Figure 2 and 4. There is a lot of red in these plots hinting that the scales might be cutting off the data. Perhaps the scales should be expanded.
Yes, this is because depolarization and backscattering of clouds is very high, comparing to aerosol. We choose such scale, to make details of aerosol more visible. So we would prefer to keep as it is.

Figure 3. Also show the smoke + pollen mixing line, since one of the selected cases references mixing of those two types.
We though to do it, but figure becomes overloaded with curves. Beside, behavior of this mixing line is quite obvious, so we think that it is not so necessary for reader.

Figures 4, 6, 912, 15, 17. It would be helpful if the curtains of intensive properties (depolarization and fluorescence capacity) had consistent scales across each of these plots, making it easier to compare one case to another.
Unfortunately, the cases are very different. In some elevated layers are considered, and in some only the PBL. So we used different scales to show the details. We would prefer to keep different scales for each episode.


References are added

Response to Reviewer 2

First of all, we would like to thank the Reviewer for reading the manuscript and useful comments.

This paper presents the potentiality of fluorescence measurements in Mie-Raman lidar systems to obtain aerosol type, with focus in biomass, dust, anthropogenic pollution and aerosols. The paper is well structured and discussions are appropriate. In general I am excited about the potential of fluorescence technique for aerosol profile characterizations. However, I agree with previous referee that authors claim the development of an algorithm and that is not straightforward from the paper. Indeed it seems an introduction with different study-cases. So my concerns prior the publication in AMT are:

- There is no mention to the physical principle of fluorescence and if fluorescence can be modeled for different aerosol particles. Maybe these models are not well developed. But if they exist, why not using them for training the model? If not, the authors should clarify this point. In summary, I miss a theoretical background for fluorescence

- The selections of the study cases are excellent, but I miss an overall conclusion that includes all your data. Why not presenting a plot that includes all data and even statistical analyses?

Minor comments

I agree with most of the comments raised by referee 1. I just would like to insist that backward-trajectories and other types of measurements (satellite, in-situ, models) would enrich the discussions. I really miss this information for the cases of Fig 1 and for the pure dust case. Also, the mention to SILAM must be clarified.

In the process of revision, the manuscript was significantly modified. We added a table, containing the particle intensive parameters for the cases considered (lidar ratios at 355 and 532 nm; depolarization ratios at 355, 532 and 1064 nm; and the backscattering and extinction Angstrom exponents). Another table provides the range of variation of particle intensive properties from different typing algorithms for the urban, smoke and dust particles. The table contains also the range of parameters variation for episodes from current study for the same aerosol types. The back-trajectory analysis is included, when the cases are analyzed. In Appendix we added four maps with SILAM pollen index, for the episodes where the presence of the pollen was revealed. Sections 3 and 4 were significantly extended and we hope, that all this improved the manuscript. Details of the manuscript revision are given in our extended response to Reviewer 1.

Reviewer is right, that at this stage we did not analyzed the fluorescence mechanisms. And this should be done at the next step of our research. We plan to increase the number of fluorescence channels, and choose of corresponding spectral intervals will demand this kind of analysis.

Statistical analysis of our observation over Lille is not done yet, but this is definitely one of our goals. And in this manuscript we tried to demonstrate that for different aerosol episode, the depolarization – fluorescence diagram allows to identify the particle type, and it also provides information about the aerosol mixture composition.

[revised manuscript text omitted]

First of all we would like to thank the reviewer for very detailed comments and useful suggestions, which helped us to improve the revised manuscript

*The manuscript describes several case studies of lidar observations where fluorescence observations combined with lidar depolarization shows significantly different properties for pollen, smoke, dust and anthropogenic aerosol. I'm excited to see the potential of these new measurements, which give completely independent and orthogonal information about aerosol particles, at single bin resolutions, significantly increasing the information available for aerosol typing. The case studies are a nice selection of different types and mixtures and interesting to see.*

*The manuscript seems to suffer from an identity problem, however. Mostly it is an illustrative set of cases studies that demonstrate differences in the two-dimensional space of fluorescence capacity and particle depolarization. It includes nice analysis of some mixtures of types as well.  However, the paper claims to be an algorithm description paper, and for that purpose, analysis of a few hand-selected case studies really isn't sufficient, and the mixture analysis doesn't exactly fit, because it is not part of the algorithm. Apparently in consequence of this uncertainty about the desired focus of the paper, some aspects of the paper seem superficial, or rather, inconsistent in depth. The inferences in the paper about the types seem very reasonable, but many are not backed up by any independent information or compared with other methods of classification, which should be done to demonstrate the validity of the new algorithm, particularly if this is the algorithm description paper. Also there's insufficient information about how the thresholds in the algorithm were chosen. In the analysis of the case studies, there should be a consistent effort to include complimentary information to validate the case identifications using  other measurements (in situ or other lidar measurements that reveal type) and backtrajectories.  And if a major focus of the paper is to showcase the performance of a new (and better) classification algorithm, then the results should be shown on a bulk of data in addition to the case studies, and comparisons with other classification methods should be made and discussed.*

The goal of this manuscript is to demonstrate that the fluorescence – depolarization diagram allows to separate different types of aerosol and provides new independent information on aerosol type, which can be used in classification schemes. The reviewer is right, at current stage of research it is not appropriate to call it "algorithm", so we escape this term in the revised manuscript.

In the revised manuscript we tried to follow the reviewer recommendations. We added a table, containing the particle intensive parameters for the cases considered (lidar ratios at 355 and 532 nm; depolarization ratios  at 355, 532 and 1064 nm; and the backscattering and extinction Angstrom exponents). Another table provides the range of variation of particle intensive properties from different typing algorithms for the urban, smoke and dust particles. The table contains also the range of parameters variation for episodes from current study for the same aerosol types.

The back-trajectory analysis is included.

In Appendix we added four maps with SILAM pollen index, for the episodes where the presence of the pollen was revealed. We hope, that all this improved the manuscript.

Specific comments:

*L24. "and their mixtures".  The mixture analysis is an interesting part of the paper, and apparently new compared to the authors' other papers, but it appears it's not really part of the classification algorithm, in the sense that mixture analysis can only be done on a case-by-case basis. Any discussion about that? This could be clarified in the abstract.  Also, the mixture analysis is not even mentioned in the introduction.  Discussing it there would help to clarify the novel aspects of the paper.*

The mixture analysis is an important but in the manuscript presented we just identify the main mixture components, based on the patterns in depolarization – fluorescence diagram. Quantification of the mixture composition is the next step in our research and corresponding algorithm is in preparation at the moment. We removed from Abstract the mentioning of mixture analysis.

*L73-75. I very much agree that adding independent aerosol information will improve classification, but this specific point is unconvincing. Yes, the variables used for classification so far have variability within types but there's nothing to suggest that this won't also be true for fluorescence capacity, is there? So, I'm not sure this is exactly the right motivation.*

The advantage of fluorescence is strong variation of fluorescence capacity between some aerosol types. For example, $G_F$ of smoke can up to one order higher, comparing to urban aerosol, allowing to separate these particles. So we think, that synergy of existing algorithms with fluorescence measurements should improve identification. Another important advantage is that $G_F$ and depolarization can be derived with high spatio – temporal resolution, so almost single pixel typing becomes possible.

*L105. Good point that the resolution is higher since fluorescence capacity can be calculated using data at a single bin, unlike extinction or other quantities related to extinction. This seems particularly useful for Raman measurements.*

Yes.

*L105-107. Veselovskii et al. 2021a is referenced extensively in the introduction, including to say that it already demonstrates the ability of the 2-d measurement space to separate all the aerosol types. I couldn't follow how the purpose and scope of this paper is different from 2021a.*

In that paper we just formulated the idea and plotted averaged data for several observations on the depolarization – fluorescence diagram. In this manuscript we evaluate the aerosol type mask with almost singe pixel resolution. Corresponding paragraph is added to the revised manuscript.

*L183-193. Calculation of the backscatter coefficient using a calibration constant sounds so straightforward, that I didn't realize that it hadn't been done before. This is great. It's good to see a relatively straightforward innovation discovered and put into practice that will produce a significant amount of additional retrievals, in profiles when the reference height is not accessible to the lidar.*

We are very pleased, that Reviewer liked our approach

*L231-232. Add an earlier reference for spectral dependence of the depolarization ratio, Burton et al. 2015.*
Added

*L240-241. Since line 223 just said that Veselovskii et al. 2021a already demonstrated that the two dimensional diagram can separate types, is the part about mixtures the main purpose of this manuscript? If so, the abstract and intro should make that clearer and the examples should be chosen to align with that purpose.*

We modified Introduction, to show that the main goal is to provide aerosol type mask with high spatio-temporal resolution. The patterns at $\delta_{532}$-$G_F$ diagram help to identify the mixture, but at current stage we can not characterize it quantitatively.

*L248-249. Burton et al. (2012) or Burton et al. (2013), referenced elsewhere in the manuscript, is an earlier lidar aerosol classification methodology with depolarization ratio ranges listed for common types.* Added

*L247. "The ranges are based on results obtained in LOA". The algorithm is a simple thresholding method in two dimensions, so the ranges are the single most important aspect of the algorithm description. This statement is much too vague to support and explain how the ranges were derived, and I'm eager to know more. What results? From cases published in other publications? From a completely independent subset of cases than the results shown in this manuscript? Are the results only inferences from the lidar measurements of depolarization and fluorescence capacity, or do they include other coincident measurements that provide stronger evidence for the type identifications? Is there a set of training cases that are classified using other external measurements and/or source information? Are the cases shown in this paper the training cases or are they independent cases that demonstrate the validation of the algorithm? All this should be part of the methodology discussion.*

We agree with reviewer and completely modified that section. We added:

**"Dust**. The depolarization ratio $\delta_{532}$ of Saharan dust near the source regions is up to 35% (Veselovskii et al., 2020a), but after transportation and mixing with local aerosol $\delta_{532}$ can be as low as 20% (Rittmeister et al., 2017). In many studies, the dust with decreased depolarization ratio is classified as "polluted dust" (e.g. Burton et al., 2012, 2013). At a moment, we do not introduce the discrimination between the two subtypes and mark as "dust" the particles with $20\% < \delta_{532} < 35\%$, and $0.1 \times 10^{-4} < G_F < 0.5 \times 10^{-4}$.

**Smoke**. In 2021-2022 we regular observed over Lille the smoke layers originated from Californian and Canadian forest fires (Hu et al., 2021). The particle depolarization and fluorescence capacity of transported smoke changed from episode to episode and for classification we choose the ranges $2\% < \delta_{532} < 10\%$, $2 \times 10^{-4} < G_F < 6 \times 10^{-4}$. At this stage we do not discriminate "fresh" and "aged" smoke, and the range of $\delta_{532}$ variation is similar to the one, used in classification of Burton et al. (2012).

**Pollen**. The pollen over north of France is usually mixed with other aerosols, and the particles, which we mark as "pollen" are actually the mixtures. Depolarization ratio of clean pollen varies strongly for different taxa. For birch pollen, Cao et al. (2010) reported $\delta_{532} = 33\%$, and in the measurements over Finland during birch pollination (Bohlmann et al., 2019), observed values of $\delta_{532}$ up to 26%. The observations over Lille during pollen season (Veselovskii et al., 2021a) rarely revealed values $\delta_{532}$ exceeding 20%. Based on that observations, we type as "pollen" the particles mixtures with $15\% < \delta_{532} < 30\%$, and $0.8 \times 10^{-4} < G_F < 3.0 \times 10^{-4}$.

**Urban**. This type of aerosol includes a variety of particle types (e.g. sulfates, soot) and its parameters may depend on the relative humidity. Based on our measurements inside the boundary layer, for classification we choose the ranges $1\% < \delta_{532} < 8\%$, and $0.1 \times 10^{-4} < G_F < 0.8 \times 10^{-}$

[4]. Similar range for $\delta_{532}$ is used in classification of Burton et al. (2012). Urban and smoke particles both have a low depolarization, but the fluorescence capacity of smoke is almost one order higher, so these particles can be reliably discriminated.

**Ice and water clouds**. Both types of the clouds have low fluorescence capacity $G_F$ <0.01×10$^{-4}$. However, the ice clouds are usually observed at the heights, where fluorescence signal is low and can not be used for classification. Thus above ~8 km the ice cloud are identified by high depolarization ratio $\delta_{532}$>40%. Depolarization ratio of the liquid water clouds is usually affected by the effects of the multiple scattering, so for their identification we use $\delta_{532}$<5%."

*Figure 3. The mixing lines all go through the box that's marked "pollen". This highlights the unavoidable weakness of typing with just two dimensions. Presumably, anything that falls within this box needs context to distinguish between pollen, a pollen mixture, or a smoke-dust mixture that has nothing to do with pollen. Identification by context (particularly where supporting measurements are available) is fine for the purpose of case studies, but there must be significant potential for misidentification in the automated algorithm, I suppose. It would be good to discuss weaknesses as well as strengths of the approach.*

Yes, aerosols are always the mixtures. So this problem is attributed not only to the presented, but also to all existing classification algorithms. Next step in our research is the increase of the number of parameters used and quantifications of mixture components.
It is true, that dust – smoke mixture, considered just at one point at depolarization – fluorescence diagram can be recognized as pollen. This is why it is important to consider all the data obtained during the session. We tried to show in this manuscript that the single pixel data for different mixtures provide different patterns, as shown in Fig.3. In our analysis we always observed this kind of patterns, and it helps to get idea about mixture composition.

*L268. Clouds are also shown in the aerosol typing masks and line 308 mentions both ice and water droplets, so the thresholds values for ice and water droplets should also be included in Table 1.*
*Figs 4,5. It's confusing that the ice cloud is only partially included in this example. It's shown in the type mask, but not discussed, and it's not shown in the scatter plot in Fig 5a. It's included in Fig 4, but apparently off-scale. The authors should decide whether they want to include the cloud in their analysis and discussion or not. If not, cut off the plots at an altitude below the cloud. If so, rescale Figure 4, include it in Fig 5 and add discussion about cloud.*

The parameters for ice and water particles are added to the Table 1. The ice clouds, however, are normally observed at high altitudes, where fluorescence signal is very weak, so corresponding points at depolarization – fluorescence diagram demonstrate strong scattering. Usually we identified the ice crystals from depolarization measurements only, and this is why we don't show them in Fig.5a. Corresponding comment is added to the revised manuscript.

*Figure 5 and similar figures. What's the purpose of the boxes and cross-hairs in the fluorescence vs. depolarization diagrams? The boxes would probably be more useful to readers if they were all the same, and used the values from Table 1. That way, we can see visually how the identified types fall into the broad category already established. I can guess that the crosshairs represent the mean and (probably) standard deviation of identified pure types, but those aren't discussed anywhere in the paper.*

In our revised manuscript the boxes correspond to Table 1. The crosses show uncertainty of our measurements, due to statistical errors and uncertainty of calibration. Corresponding comment is added to revised manuscript.

*L315-321. The explanation of the smoothing procedure is missing something. Z is a number, but the classification IDs are not numbers that can be added and weighted, but just labels. How are the classifications convolved with Z? Just guessing, I suppose the fluorescence capacity and depolarization ratio are what's averaged using the Z-weightings, and then the classification is done on these smoothed measurements instead? Please clarify in the text.*

To make it more clear, we modified corresponding section in the revised manuscript and extended description.
Briefly:
We construct several 'raw' matrices with dimensions equal to primary data matrices (one matrix for each aerosol type (dust, pollen, etc)). If at the first stage some single pixel data point (i,j) is classified as, e.g., pollen, the corresponding value in the 'pollen' matrix is set to 1, otherwise it is set to 0. Then each of these matrices is separately convoluted with the Gauss kernel Z. And, after the convolution, the values for each pixel data (i,j) are being compared. If, e.g., the 'dust' matrix (after the convolution) contains maximal value at the point (i,j) among all the matrices (after the convolution), then the point (i,j) is finally classified as 'dust'.

*L339 and 341 and elsewhere. I'd suggest avoiding describing values as "typical" and expand the description to be more specific. For instance, perhaps this is within the ranges seen in your previous publications and/or other publications for cases that have been identified as smoke and urban based on independent data? "Typical" is a bit dangerous, in that it implies a generality that is not established after only a few handfuls of case studies, particularly since the case study identifications seem to mostly be rather dependent on expectations about the typical values. Statements like this unfortunately seem to be quoted and referenced repeatedly so that they become ingrained without becoming better supported. After all, we now know that it is quite common for smoke (in the upper troposphere and stratosphere) to have depolarization values that are much larger than this, and previously published ranges of depolarization for urban aerosol also include significantly larger depolarization values than this.*

Agree. We tried to follow this recommendation in revised manuscript
It is true, that aged smoke depolarization ratio at 532 nm in stratosphere can be as high as ~20%. We should mention also, that at 1064 nm the depolarization ratio of smoke in our measurements (even in upper troposphere) never exceeded 5%. This is one more reason to include this depolarization ratio in typing scheme at next stage.

*L347. Says that the fluorescence capacity can decrease as a function of relative humidity, explaining a range of variables. Why does it produce variability rather than reducing the fluorescence capacity uniformly?*
The water uptake increases the particle backscattering, but does not change the fluorescence. As a result, the fluorescence capacity decreases. The RH, changes with height, which can lead to increase of single pixel data scattering inside the cluster.

*L361-367 and Figure 6-7. I agree that the shape of the curve in Figure 7a is very striking and reminiscent of a mixing line. However, I also just read in the previous section that fluorescence capacity is strongly impacted by relative humidity, making me wonder quantitatively how much impact RH has, compared to the impact of mixing. Is there a model (theoretical or empirical) of G_F dependence on relative humidity? The RH profile should be added to Figure 8 (and all the other profile figures). Another aspect that puzzles*

*and surprises me is the increased G_F specifically in parts of the curtain where the backscatter is lower. This hints that the variation in G_F might be quite strongly related to RH; alternately that the pollen is more diffuse and widespread than the urban aerosol, which I think would be unusual. A curtain of RH (perhaps from MERRA-2 since there is insufficient sonde data to produce a curtain) and/or backtrajectories might help make the scenario more clear.*

Unfortunately, we had no collocated RH measurements. The sonde measurements in UK show that RH increased from 40% to 70% with height. The value of the fluorescence capacity changed for one order of magnitude, and such strong change in $G_F$ can not be explained by the particle hygroscopic growth. For example, from the recent publication of Sicard et al., increase of $\beta_{532}$ in this RH range for urban aerosol is below factor 1.5. (Sicard, M., Fortunato dos Santos Oliveira, D. C., Muñoz-Porcar, C., Gil-Díaz, C., Comerón, A., Rodríguez-Gómez, A., and Dios Otín, F.: Measurement Report: Spectral and statistical analysis of aerosol hygroscopic growth from multi-wavelength lidar measurements in Barcelona, Spain, Atmos. Chem. Phys. 22, 7681–7697, 2022). Corresponding comment is added to revised manuscript.

The hygroscopic growth can contribute to the backscattering near the PBL top. However, at low altitudes RH is about 40%, so increase of $G_F$ is probably due to decrease of urban particles contribution to the total backscattering (thus pollen contribution becomes more visible). We tried to use MERRA-2 data, but at low altitudes the modeled parameters differed strongly from observations.

*L368-369. It's good that 1064 nm depolarization is included here, because in general, the more data shown, the better the patterns can be understood. However, the text highlights larger values of 1064 nm depolarization to support the inference of pollen, but that's also true for urban aerosol (e.g. Burton et al. 2012). Then "both depolarization ratios decrease with height" as the pollen concentration decreases (L372), but 1064 continues to be larger than 532, so again this is not definitive. Any further comment about this?*

Yes, urban aerosol may also have $\delta_{1064}$ exceeding $\delta_{532}$. But absolute values of depolarization for pollen are significantly higher. So when at low altitudes we observe high $G_F$, and high depolarization, the observed $\delta_{1064} > \delta_{532}$ corroborates presence of pollen.

*This case and the first case were also included in earlier publications by the same authors. The papers make different analyses of them, so that's fine, but does this mean they also contributed information relevant to producing the ranges used in the algorithm? If so, they are not such good examples to illustrate the performance of the typing algorithm.*

The typing is performed on a base of $G_F$-$\delta_{532}$ measurements only. We used these examples, because the aerosol origin was analyzed in our previous publications. Besides, measurements on 30 May 2020 demonstrate very characteristic pattern for urban – pollen mixture.

The vertical profiles of particle parameters for 30 May were presented in our recent paper, so we decided to exclude Fig.8 from revised manuscript. We just provide the reference.

*Figure 8 L 715. Why were the profiles created for 21:00-23:00 instead of a later time where the curtain shows pollen at lower altitudes and mixing is discussed? Is this a mistake?*

Sorry, this was mistake.

*L376-377. I'm not finding the explanation for the lack of variability in the backscatter angstrom exponent to be very convincing. It appears to be saying that the urban particles are growing due to humidification exactly in balance with the effective dry particle size decreasing due to less pollen? (if so, this needs support).*
*Perhaps some quantitative modeling would help. How small of a backscatter Angstrom exponent would be expected for high concentration of pollen, and just how much contribution to the backscatter is there (based on the mixing model) and how much change in Angstrom would you therefore expect? What confuses me is that the fluorescence capacity also mixes linearly according to the backscatter partition, so if there was really too little backscatter contribution to be noticeable, wouldn't that also mean there would be little variation in G_F as well?*

In revised manuscript, this section was completely modified. We agree with reviewer, that behavior of backscatter Angstrom 532/1064 is puzzling. However, the observation presented, could be strongly influenced by hygroscopic growth, which decreases both depolarization and the fluorescence capacity. The backscattering Angstrom exponent strongly (and in complicated way) depends on refractive index, particle size and particle shape. The modeling of the BAE for different mixture compositions is important, but it is out of scope of this research. Just want to mention, that that in publication of Bohlmann et al. (2019) the BAE (at depolarization ratio ~20%) is about 1.0. Which is quite high value and pollen content over Finland is significantly higher than over Lille. So this aspect needs additional research and additional measurements during strong pollen episodes.

*L392-393. Unfortunately, the SILAM website only provides current forecast data, so please make the relevant data available as a supplement or shown in a figure. Also, what kind of pollen was it?*

In situ measurements at the roof of the building demonstrate presence of significant amount of grass pollen. We added to the revised manuscript (as Appendix) the SILAM maps for four episodes, when presence of pollen was assumed.

L418-419. I'm not quite clear on what the author's intent is here. Is this saying that the algorithm misclassified a mixture as pure urban, or that the mixture only occurs where the classification puts it, but that the two urban layers have quite a lot of difference between them? It would be very helpful (in this case and others) to mark the points in the scatterplots according to the classification result or altitude. I would like to see exactly where the two layers classified as "urban" fall on the apparent mixing line. I think it's interesting that the two layers marked urban have different spectral dependence of depolarization. Backtrajectories would be helpful for this case too, to help understand why the two layers of urban aerosol might have different properties.

To make presentation more clear, we significantly modified this section. First of all, in depolarization – fluorescence diagram in Fig.12 we show the points related to the upper and lower layers by different colors. Back trajectories analysis shows that air masses in both layers are transported from England. So this is probably pollution. Points related to the upper layer are inside the range for 'urban' aerosol. Points in the lower layer, are partly outside of this range, so the aerosol type is undefined. We assume that this is the mixture of urban and pollen particles, because we have particles with high depolarization and fluorescence capacity (still not high enough to be classified as "pollen"). This mixture is marked by grey color and it is located below 750 m. The maps with SILAM pollen index are added to the revised manuscript as Appendix. On the midnight of 10-11 April 2020 the pollen loading is modeled by SILAM as moderate. Thus yes, properties of layers are different. Upper layer is urban, while in lower layer below 1 km the urban particles are mixed with pollen.

L420. "typical for urban-pollen mixture". Actually the mixing curve is significantly to the left of the curve in Figure 3, suggesting that the pure pollen in this mixture is not "typical" compared to the ranges given in the table, but is more of an edge case with relatively low fluorescence capacity.

Yes, fluorescence capacity is lower than usual, so this not pure pollen. We added corresponding comment to the text.

L430-436.  This is a very nice case to demonstrate contrast in fluorescence between different types.  But the type identification is entirely made by inference using the two classification dimensions without any other support such as in situ measurements, backtrajectories, or other lidar-measured quantities like 1064 nm depolarization, lidar ratio or angstrom exponents. It's great that two measurements used for the classification appear to give the ability to make these separations, but for such a key demonstration I think the case studies need to be very well supported.  In general I suggest bolstering the verification of the identifications for all the cases (not just this one) by including all relevant data.  I mean specifically, first of all, other lidar quantities that have been used in previous classification methodologies, including especially lidar ratios, and also 1064 nm depolarization and angstrom exponents for all cases.  Also include RH, backtrajectories and any coincident in situ measurements (especially pollen) for all cases.

In the revised manuscript we added the Table 2, with main intensive particle parameters for all episode considered. The section is modified: we added backtrajectories and analysis of the intensive particle parameters. We have added also Table 3, which compares our observed intensive parameters for dust, smoke, urban with parameters used in existing typing algorithms.

L445 and Figs 15 and 16. The suggested mixing between layers doesn't look convincing.  On the fluorescence vs. depolarization diagram, these intermediate points don't follow a nice mixing line like the other mixing cases, and the boundaries in the measurement curtains appear quite crisp. Could these points be artifacts of the smoothing instead?

Yes, at high gradients of backscattering, smoothing sometimes can provide oscillation. We reprocessed this case with decreased smoothing. The threshold value of $\beta_{532}$ was increased up to 0.3 Mm-1sr-1. Now it is better.

Fig 15. The depolarization especially and perhaps also the fluorescence capacity (outside of the smoke plume) seems to be anti-correlated with backscatter, including in regions that seem unlikely to be pollen-dominated (such as the minimum between the smoke and urban layers). Particulate depolarization is especially susceptible to systematic error, particularly overestimation, at low values of backscatter (Freudenthaler et al. 2009, Burton et al. 2015).  Have you done a systematic uncertainty calculation? (Also this is another case where color coding of the scatterplot by altitude would be useful).

Yes, calculation of depolarization at low $\beta_{532}$ can lead to enhanced uncertainty, especially when high gradients of $\beta_{532}$ present. In reprocessed data we increased threshold value of $\beta_{532}$ up to 0.3 Mm-1sr-1. Oscillations decreased. The same is true for fluorescence capacity.
We estimate uncertainty of our depolarization calibration to be below 15%.

Fig 15-18. Include the data for depolarization and angstrom exponent (and RH) for these cases also.

In the revised manuscript we have added Figures 16, 19 with vertical profiles for these episodes.

L455 "G_F increased ... probably due to the mixing with local pollution".  Does this make sense?  Nothing prior to this in the manuscript suggests that urban pollution has significant fluorescence capacity.  Also, on the scatter plot on Figure 18, there's no suggestion that the higher values of G_F in the dust cluster are correlated with depolarization in any way; that is, they are not following any mixing line. What evidence is there that this is not simply normal variability within dust?  Table 1 shows dust can have G_F up to 0.5.  Why not 0.6?  Also, could some of this variability be correlated with RH?

Dust may have very low fluorescence capacity (0.1*10^-4), while urban particles for some episodes had $G_F$ of 0.8*10^-4, or even higher. Thus mixing of dust with pollutions, in principle, can increase the capacity. But reviewer is right, for case presented, the depolarization ratio did not change significantly with height, while capacity strongly decreased in the center of the layer. It can be variation of dust composition (and so the absorption) through the layer. Unfortunately, at this stage we can not make definite conclusion. Corresponding section is strongly modified in revised manuscript.

For the available dust episodes the fluorescence capacity was mainly below 0.5*10^-4. This is why we used it in Table 1. We may reconsider this range, when more data will be available.

Normally properties of dust are not very sensitive to RH. Increase of RH can only decrease the capacity. However at a moment we don't have collocated RH measurements, so unable to make quantitative conclusions about RH influence.

L485. "during Spring-Autumn seasons". It would be helpful to show a timeseries demonstrating that the pollen signature (elevated depolarization and fluorescence capacity) does NOT occur in winter.

We agree, that this would be useful, but it is beyond the scope of this manuscript. Seasonal variation of aerosol composition over Lille will be the topic of separate research.

Typographical or wording:
L19. What is meant by "single" in "first single version of the algorithm". I suggest delete "single" or reword.
Corrected

L18 and L24. Change particle's to particle.
Corrected

L92. Be specific about which wavelength here.
Done

L247. Define LOA.
Done

L270-281. There should be some discussion or at least references to other analyses of mixtures of aerosols that derive similar equations (especially Eq. 7), e.g. Sugimoto and Lee 2006, Gross et al. 2011, Gasteiger et al. 2011, Tesche et al. 2009, Burton et al. 2014.

The references are added. Derivation of Eq.7 looks very straightforward, so probably no explanations are needed.

L280. Eq. 8. It probably would be good to remind the reader that fluorescence capacity and backscatter in this equation refer to particular wavelengths.
Done

L282. "We assume". I think this is meant to refer only to the demonstration in Figure 3, not a general assertion. If true, perhaps swap the first two sentences of the paragraph to make it less likely to be misread. As mentioned in the introduction, the quantities have a lot of variability even within types, so assuming single values wouldn't be well supported.

Done

L300. "the height resolution is 7.5 m".  Is that really the resolution or only the grid spacing?  That is, taking the detectors into account, are measurements at adjacent vertical grid points independent?

Yes, this is bin resolution of our detection electronics, and in many cases this resolution was used to calculate the particle properties. However, for elevated layers the fluorescence signal was splined. For typing, the Gaussian smoothing procedure was used. Thus ultimate resolution was about 60 m for height and less than 10 minutes for time.

L342. spell out FBC
The section was modified

L344.  Add a reference to the reminder.  (I think it is Veselovskii 2020?)
Done

L445.  Typo in "0.2-0.3"
Corrected

L663-664. It would be helpful to add "using the reference height as Ansmann et al. 1992 (green) or the calibration constant as in Eq 5. (magenta)". (I read figure captions before the text, so having a bit more detail in the captions is very helpful)
Done

L708. Please add clarification to the caption whether the scatter plot shows data for the entire time period shown in the curtain or only the subset that's included in the profile plots of Figure 8.
This is for entire time period. Added to caption.

Figure 2 and 4.  There is a lot of red in these plots hinting that the scales might be cutting off the data.  Perhaps the scales should be expanded.
Yes, this is because depolarization and backscattering of clouds  is very high, comparing to aerosol. We choose such scale, to make details of aerosol more visible. So we would prefer to keep as it is.

Figure 3.  Also show the smoke + pollen mixing line, since one of the selected cases references mixing of those two types.
We though to do it, but figure becomes overloaded with curves. Beside, behavior of this mixing line is quite obvious, so we think that it is not so necessary for reader.

Figures 4, 6, 912, 15, 17. It would be helpful if the curtains of intensive properties (depolarization and fluorescence capacity) had consistent scales across each of these plots, making it easier to compare one case to another.
Unfortunately, the cases are very different. In some elevated layers are considered, and in some only the PBL. So we used different scales to show the details. We would prefer to keep different scales for each episode.

References are added